# SELECTIVITY CONSIDERED HARMFUL: EVALUATING THE CAUSAL IMPACT OF CLASS SELECTIVITY IN DNNS

**Matthew L. Leavitt,**[*] **Ari S. Morcos**
Facebook AI Research
Menlo Park, CA, USA
`{ito,arimorcos}@fb.com`

## ABSTRACT

The properties of individual neurons are often analyzed in order to understand the biological and artificial neural networks in which they're embedded. Class selectivity—typically defined as how different a neuron's responses are across different classes of stimuli or data samples—is commonly used for this purpose. However, it remains an open question whether it is necessary and/or sufficient for deep neural networks (DNNs) to learn class selectivity in individual units. We investigated the causal impact of class selectivity on network function by directly regularizing for or against class selectivity. Using this regularizer to reduce class selectivity across units in convolutional neural networks increased test accuracy by over 2% in ResNet18 and 1% in ResNet50 trained on Tiny ImageNet. For ResNet20 trained on CIFAR10 we could reduce class selectivity by a factor of 2.5 with no impact on test accuracy, and reduce it nearly to zero with only a small (∼2%) drop in test accuracy. In contrast, regularizing to increase class selectivity significantly decreased test accuracy across all models and datasets. These results indicate that class selectivity in individual units is neither sufficient nor strictly necessary, and can even impair DNN performance. They also encourage caution when focusing on the properties of single units as representative of the mechanisms by which DNNs function.

## 1 INTRODUCTION

Our ability to understand deep learning systems lags considerably behind our ability to obtain practical outcomes with them. A breadth of approaches have been developed in attempts to better understand deep learning systems and render them more comprehensible to humans (Yosinski et al., 2015; Bau et al., 2017; Olah et al., 2018; Hooker et al., 2019). Many of these approaches examine the properties of single neurons and treat them as representative of the networks in which they're embedded (Erhan et al., 2009; Zeiler and Fergus, 2014; Karpathy et al., 2016; Amjad et al., 2018; Lillian et al., 2018; Dhamdhere et al., 2019; Olah et al., 2020).

The selectivity of individual units (i.e. the variability in a neuron's responses across data classes or dimensions) is one property that has been of particular interest to researchers trying to better understand deep neural networks (DNNs) (Zhou et al., 2015; Olah et al., 2017; Morcos et al., 2018b; Zhou et al., 2018; Meyes et al., 2019; Na et al., 2019; Zhou et al., 2019; Rafegas et al., 2019; Bau et al., 2020). This focus on individual neurons makes intuitive sense, as the tractable, semantic nature of selectivity is extremely alluring; some measure of selectivity in individual units is often provided as an explanation of "what" a network is "doing". One notable study highlighted a neuron selective for sentiment in an LSTM network trained on a word prediction task (Radford et al., 2017). Another attributed visualizable, semantic features to the activity of individual neurons across GoogLeNet trained on ImageNet (Olah et al., 2017). Both of these examples influenced many subsequent studies, demonstrating the widespread, intuitive appeal of "selectivity" (Amjad et al., 2018; Meyes et al., 2019; Morcos et al., 2018b; Zhou et al., 2015; 2018; Bau et al., 2017; Karpathy et al., 2016; Na et al., 2019; Radford et al., 2017; Rafegas et al., 2019; Morcos et al., 2018b; Olah et al., 2017; 2018; 2020).

---

[*]Work performed as part of the Facebook AI Residency

Finding intuitive ways of representing the workings of DNNs is essential for making them understandable and accountable, but we must ensure that our approaches are based on meaningful properties of the system. Recent studies have begun to address this issue by investigating the relationships between selectivity and measures of network function such as generalization and robustness to perturbation (Morcos et al., 2018b; Zhou et al., 2018; Dalvi et al., 2019). Selectivity has also been used as the basis for targeted modulation of neural network function through individual units (Bau et al., 2019a;b).

However there is also growing evidence from experiments in both deep learning (Fong and Vedaldi, 2018; Morcos et al., 2018b; Gale et al., 2019; Donnelly and Roegiest, 2019) and neuroscience (Leavitt et al., 2017; Zylberberg, 2018; Insanally et al., 2019) that single unit selectivity may not be as important as once thought. Previous studies examining the functional role of selectivity in DNNs have often measured how selectivity mediates the effects of ablating single units, or used indirect, correlational approaches that modulate selectivity indirectly (e.g. batch norm) (Morcos et al., 2018b; Zhou et al., 2018; Lillian et al., 2018; Meyes et al., 2019; Kanda et al., 2020). But single unit ablation in trained networks has two critical limitations: it cannot address whether the *presence* of selectivity is beneficial, nor whether networks *need to learn* selectivity to function properly. It can only address the effect of removing a neuron from a network whose training process assumed the presence of that neuron. And even then, the observed effect might be misleading. For example, a property that is critical to network function may be replicated across multiple neurons. This redundancy means that ablating any one of these neurons would show little effect, and could thus lead to the erroneous conclusion that the examined property has little impact on network function.

We were motivated by these issues to pursue a series of experiments investigating the causal importance of class selectivity in artificial neural networks. To do so, we introduced a term to the loss function that allows us to directly regularize for or against class selectivity, giving us a single knob to control class selectivity in the network. The selectivity regularizer sidesteps the limitations of single unit ablation and other indirect techniques, allowing us to conduct a series of experiments evaluating the causal impact of class selectivity on DNN performance. Our findings are as follows:

- Performance can be improved by reducing class selectivity, suggesting that naturally-learned levels of class selectivity can be detrimental. Reducing class selectivity could improve test accuracy by over 2% in ResNet18 and 1% in ResNet50 trained on Tiny ImageNet.
- Even when class selectivity isn't detrimental to network function, it remains largely unnecessary. We  reduced the mean class selectivity of units in ResNet20 trained on CIFAR10 by a factor of $\sim$2.5 with no impact on test accuracy, and by a factor of $\sim$20—nearly to a mean of 0—with only a 2% change in test accuracy.
- Our regularizer does not simply cause networks to preserve class-selectivity by rotating it off of unit-aligned axes (i.e. by distributing selectivity linearly across units), but rather seems to suppress selectivity more generally, even when optimizing for high-selectivity basis sets . This demonstrates the viability of low-selectivity representations distributed *across* units.
- We show that regularizing to increase class selectivity, even by small amounts, has significant negative effects on performance. Trained networks seem to be perched precariously at a performance cliff with regard to class selectivity. These results indicate that the levels of class selectivity learned by individual units in the absence of explicit regularization are at the limit of what will impair the network.

Our findings collectively demonstrate that class selectivity in individual units is neither necessary nor sufficient for convolutional neural networks (CNNs) to perform image classification tasks, and in some cases can actually be detrimental. This alludes to the possibility of class selectivity regularization as a technique for improving CNN performance. More generally, our results encourage caution when focusing on the properties of single units as representative of the mechanisms by which CNNs function, and emphasize the importance of analyses that examine properties across neurons (i.e. distributed representations). Most importantly, our results are a reminder to verify that the properties we *do* focus on are actually relevant to CNN function.

## 2 RELATED WORK

### 2.1 SELECTIVITY IN DEEP LEARNING

Examining some form of selectivity in individual units constitutes the bedrock of many approaches to understanding DNNs. Sometimes the goal is simply to visualize selectivity, which has been pursued

using a breadth of methods. These include identifying the input sample(s) (e.g. images) or sample subregions that maximally activate a given neuron (Zhou et al., 2015; Rafegas et al., 2019), and numerous optimization-based techniques for generating samples that maximize unit activations (Erhan et al., 2009; Zeiler and Fergus, 2014; Simonyan et al., 2014; Yosinski et al., 2015; Nguyen et al., 2016; Olah et al., 2017; 2018). While the different methods for quantifying single unit selectivity are often conceptually quite similar (measuring how variable are a neuron's responses across different classes of data samples), they have been applied across a broad range of contexts (Amjad et al., 2018; Meyes et al., 2019; Morcos et al., 2018b; Zhou et al., 2015; 2018; Bau et al., 2017; Karpathy et al., 2016; Na et al., 2019; Radford et al., 2017; Rafegas et al., 2019). For example, Bau et al. (2017) quantified single unit selectivity for "concepts" (as annotated by humans) in networks trained for object and scene recognition. Olah et al. (2018; 2020) have pursued a research program examining single unit selectivity as a building block for understanding DNNs. And single units in models trained to solve natural language processing tasks have been found to exhibit selectivity for syntactical and semantic features (Karpathy et al., 2016; Na et al., 2019), of which the "sentiment-selective neuron" reported by Radford et al. (2017) is a particularly recognized example.

The relationship between individual unit selectivity and various measures of DNN performance have been examined in prior studies, but the conclusions have not been concordant. Morcos et al. (2018b), using single unit ablation and other techniques, found that a network's test set generalization is negatively correlated (or uncorrelated) with the class selectivity of its units, a finding replicated by Kanda et al. (2020). In contrast, though Amjad et al. (2018) confirmed these results for single unit ablation, they also performed cumulative ablation analyses which suggested that selectivity is beneficial, suggesting that redundancy across units may make it difficult to interpret single unit ablation studies.

In a follow-up study, Zhou et al. (2018) found that ablating class-selective units impairs classification accuracy for specific classes (though interestingly, not always the same class the unit was selective for), but a compensatory increase in accuracy for other classes can often leave overall accuracy unaffected. Ukita (2018) found that orientation selectivity in individual units is correlated with generalization performance in convolutional neural networks (CNNs), and that ablating highly orientation-selective units impairs classification accuracy more than ablating units with low orientation-selectivity. But while orientation selectivity and class selectivity can both be considered types of feature selectivity, orientation selectivity is far less abstract and focuses on specific properties of the image (e.g., oriented edges) rather than semantically meaningful concepts and classes. Nevertheless, this study still demonstrates the importance of some types of selectivity.

Results are also variable for models trained on NLP tasks. Dalvi et al. (2019) found that ablating units selective for linguistic features causes greater performance deficits than ablating less-selective units, while Donnelly and Roegiest (2019) found that ablating the "sentiment neuron" of Radford et al. (2017) has equivocal effects on performance. These findings seem challenging to reconcile.

All of these studies examining class selectivity in single units are hamstrung by their reliance on single unit ablation, which could account for their conflicting results. As discussed earlier, single unit ablation can only address whether class selectivity affects performance in trained networks, and not whether individual units to *need to learn* class selectivity for optimal network function. And even then, the conclusions obtained from single neuron ablation analyses can be misleading due to redundancy across units (Amjad et al., 2018; Meyes et al., 2019).

## 2.2 SELECTIVITY IN NEUROSCIENCE

Measuring the responses of single neurons to a relevant set of stimuli has been the canonical first-order approach for understanding the nervous system (Sherrington, 1906; Adrian, 1926; Granit, 1955; Hubel and Wiesel, 1959; Barlow, 1972; Kandel et al., 2000); its application has yielded multiple Nobel Prizes (Hubel and Wiesel, 1959; 1962; Hubel, 1982; Wiesel, 1982; O'Keefe and Dostrovsky, 1971; Fyhn et al., 2004). But recent experimental findings have raised doubts about the necessity of selectivity for high-fidelity representations in neuronal populations (Leavitt et al., 2017; Insanally et al., 2019; Zylberberg, 2018), and neuroscience research seems to be moving beyond characterizing neural systems at the level of single neurons, towards population-level phenomena (Shenoy et al., 2013; Raposo et al., 2014; Fusi et al., 2016; Morcos and Harvey, 2016; Pruszynski and Zylberberg, 2019; Heeger and Mackey, 2019; Saxena and Cunningham, 2019).

Single unit selectivity-based approaches are ubiquitous in attempts to understand artificial and biological neural systems, but growing evidence has led to questions about the importance of focusing on selectivity and its role in DNN function. These factors, combined with the limitations of prior approaches, lead to the question: is class selectivity necessary and/or sufficient for DNN function?

## 3 APPROACH

Networks naturally seem to learn solutions that result in class-selective individual units (Zhou et al., 2015; Olah et al., 2017; Morcos et al., 2018b; Zhou et al., 2018; Meyes et al., 2019; Na et al., 2019; Zhou et al., 2019; Rafegas et al., 2019; Amjad et al., 2018; Meyes et al., 2019; Bau et al., 2017; Karpathy et al., 2016; Radford et al., 2017; Olah et al., 2018; 2020). We examined whether learning class-selective representations in individual units is actually necessary for networks to function properly. Motivated by the limitations of single unit ablation techniques and the indirectness of using batch norm or dropout to modulate class selectivity (e.g. Morcos et al. (2018b); Zhou et al. (2018); Lillian et al. (2018); Meyes et al. (2019)), we developed an alternative approach for examining the necessity of class selectivity for network performance. By adding a term to the loss function that serves as a regularizer to suppress (or increase) class selectivity, we demonstrate that it is possible to directly modulate the amount of class selectivity in all units in aggregate. We then used this approach as the basis for a series of experiments in which we modulated levels of class selectivity across individual units and measured the resulting effects on the network. Critically, the selectivity regularizer sidesteps the limitations of single unit ablation-based approaches, allowing us to answer otherwise-inaccessible questions such as whether single units actually need to learn class selectivity, and whether increased levels of class selectivity are beneficial.

Unless otherwise noted: all experimental results were derived from the test set with the parameters from the epoch that achieved the highest validation set accuracy over the training epochs; 20 replicates with different random seeds were run for each hyperparameter set; error bars and shaded regions denote bootstrapped 95% confidence intervals; selectivity regularization was not applied to the output (logits), nor was the output included in any of our analyses because by definition the output must be class selective in a classification task. Selectivity regularization was only applied to intermediate (hidden) layers with non-linearities.

### 3.1 MODELS AND DATASETS

Our experiments were performed on ResNet18 and ResNet50 (He et al., 2016) trained on Tiny ImageNet (Fei-Fei et al., 2015), and ResNet20 (He et al., 2016) and a VGG16-like network (Simonyan and Zisserman, 2015), both trained on CIFAR10 (Krizhevsky, 2009). Additional details about hyperparameters, data, training, and software are in Appendix A.1. We focus on ResNet18 trained on Tiny ImageNet in the main text, but results were qualitatively similar across models and datasets except where noted.

### 3.2 DEFINING CLASS SELECTIVITY

There are a breadth of approaches for quantifying class selectivity in individual units (Moody et al., 1998; Zhou et al., 2015; Li et al., 2015; Zhou et al., 2018; Gale et al., 2019). We chose the neuroscience-inspired approach of Morcos et al. (2018b) because it is similar to many widely-used metrics, easy to compute, and most importantly, differentiable (the utility of this is addressed in the next section). We also confirmed the efficacy of our regularizer on a different, non-differentiable selectivity metric (see Appendix A.13). For a single convolutional feature map (which we refer to as a "unit"), we computed the mean activation across elements of the filter map in response to a single sample, after the non-linearity. Then the class-conditional mean activation (i.e. the mean activation for each class) was calculated across all samples in the test set, and the class selectivity index ($SI$) was calculated as follows:

$$SI = \frac{\mu_{max} - \mu_{-max}}{\mu_{max} + \mu_{-max} + \epsilon} \tag{1}$$

where $\mu_{max}$ is the largest class-conditional mean activation, $\mu_{-max}$ is the mean response to the remaining (i.e. non-$\mu_{max}$) classes, and $\epsilon$ is a small value to prevent division by zero (we used $10^{-7}$) in the case of a dead unit. The selectivity index can range from 0 to 1. A unit with identical average

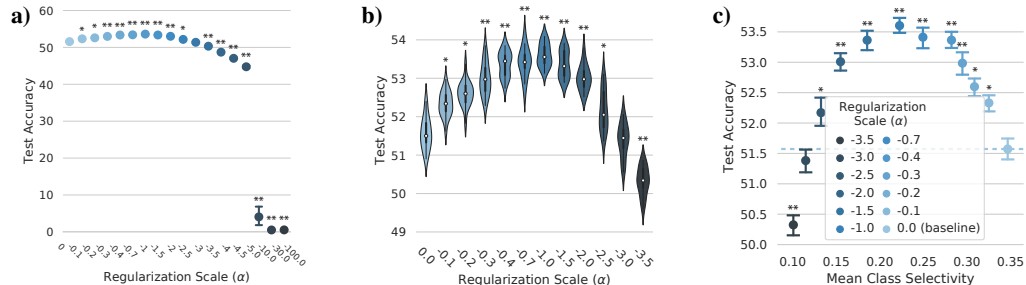

**Figure 1: Effects of reducing class selectivity on test accuracy in ResNet18 trained on Tiny Imagenet.** (**a**) Test accuracy (y-axis) as a function of regularization scale ($\alpha$, x-axis and intensity of blue). (**b**) Identical to (**a**), but for a subset of $\alpha$ values. The center of each violin plot contains a boxplot, in which the darker central lines denote the central two quartiles. (**c**) Test accuracy (y-axis) as a function of mean class selectivity (x-axis) for different values of $\alpha$. Error bars denote 95% confidence intervals. *$p < 0.01$, **$p < 5 \times 10^{-10}$ difference from $\alpha = 0$, t-test, Bonferroni-corrected. See Appendix A.4 and A.12 for ResNet20 and VGG results, respectively.

activity for all classes would have a selectivity of 0, and a unit that only responded to a single class would have a selectivity of 1.

As Morcos et al. (2018b) note, this selectivity index is not a perfect measure of information content in single units. For example, a unit with some information about many classes would have a low selectivity index. But it achieves the goal of identifying units that are class-selective in a similarly intuitive way as prior studies (Zhou et al., 2018), while also being differentiable with respect to the model parameters.

### 3.3 A SINGLE KNOB TO CONTROL CLASS SELECTIVITY

Because the class selectivity index is differentiable, we can insert it into the loss function, allowing us to directly regularize for or against class selectivity. Our loss function, which we seek to minimize, thus takes the following form:

$$loss = - \sum_c^C y_c \cdot \log(\hat{y}_c) - \alpha \mu_{SI} \tag{2}$$

The left-hand term in the loss function is the traditional cross-entropy between the softmax of the output units and the true class labels, where $c$ is the class index, $C$ is the number of classes, $y_c$ is the true class label, and $\hat{y}_c$ is the predicted class probability. We refer to the right-hand component of the loss function, $-\alpha \mu_{SI}$, as the class selectivity regularizer (or regularizer, for brevity). The regularizer consists of two terms: the selectivity term,

$$\mu_{SI} = \frac{1}{L} \sum_l^L \frac{1}{U} \sum_u^U SI_u \tag{3}$$

where $l$ is a convolutional layer, $L$ is number of layers, $u$ is a unit (i.e. feature map), $U$ is the number of units in a given layer, and $SI_u$ is the class selectivity index of unit $u$. The selectivity term of the regularizer is obtained by computing the selectivity index for each unit in a layer, then computing the mean selectivity index across units within each layer, then computing the mean selectivity index across layers. Computing the mean within layers before computing the mean across layers (as compared to computing the mean across all units in the network) mitigates the biases induced by the larger numbers of units in deeper layers. The remaining term in the regularizer is $\alpha$, the regularizer scale. The sign of $\alpha$ determines whether class selectivity is promoted or discouraged. Negative values of $\alpha$ discourage class selectivity in individual units, while positive values promote it. The magnitude of $\alpha$ controls the contribution of the selectivity term to the overall loss. $\alpha$ thus serves as a single knob with which we can modulate class selectivity across all units in the network in aggregate. During training, the class selectivity index was computed for each minibatch. For the results presented here, the class selectivity index was computed across the entire test set. We also tried restricting regularization to the first or final three layers (Appendix A.15), and warming up the class selectivity regularization over the initial training epochs (Appendix A.16), all of which yielded qualitatively similar results.

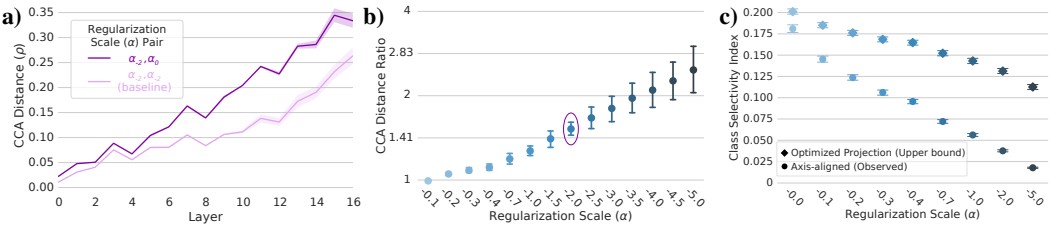

**Figure 2: Checking for off-axis selectivity.** (**a**) Mean CCA distance ($\rho$, y-axis) as a function of layer (x-axis) between pairs of replicate ResNet18 networks (see Section 4.2 or Appendix A.3.2) trained with $\alpha = -2$ (i.e. $\rho(\alpha_{-2}, \alpha_{-2})$; light purple), and between pairs of networks trained with $\alpha = -2$ and $\alpha = 0$ (i.e. $\rho(\alpha_{-2}, \alpha_0)$; dark purple). (**b**) For each layer, we compute the ratio of $\rho(\alpha_{-2}, \alpha_0) : \rho(\alpha_{-1}, \alpha_{-2})$, which we refer to as the CCA distance ratio. We then plot the mean CCA distance ratio across layers (y-axis) as a function of $\alpha$ (x-axis, intensity of blue). Example from panel **a** ($\alpha = -2$) circled in purple. $p < 1.3 \times 10^{-5}$, paired t-test, for all $\alpha$ except -0.1. (**c**) Mean class selectivity (y-axis) as a function of regularization scale ($\alpha$; x-axis) for ResNet18 trained on Tiny ImageNet. Diamond-shaped data points denote the upper bound on class selectivity for a linear projection of activations (see Section 4.2 or Appendix A.7), while circular points denote the amount of axis-aligned class selectivity for the corresponding values of $\alpha$. Error bars or shaded region = 95% confidence intervals. ResNet20 results are in Appendix A.6 (CCA) and A.7 (selectivity upper bound).

## 4 RESULTS

### 4.1 TEST ACCURACY IS IMPROVED OR UNAFFECTED BY REDUCING CLASS SELECTIVITY

Prior research has yielded equivocal results regarding the importance of class selectivity in individual units. We sidestepped the limitations of previous approaches by regularizing against selectivity directly in the loss function through the addition of the selectivity term (see Approach 3.3), giving us a knob with which to causally manipulate class selectivity. We first verified that the regularizer works as intended (Figure A1). Indeed, class selectivity across units in a network decreases as $\alpha$ becomes more negative. We also confirmed that our class selectivity regularizer has similar effects when measured using a different class selectivity metric and mutual information (see Appendix A.13), and when regularizing to control $d'$—a measure of class discriminability—in individual units (Appendix A.14). The consistency of our observations across metrics of selectivity indicates that our results are not unique to the metric used in our regularizer. The regularizer thus allows us to to examine the causal impact of class selectivity on test accuracy.

Regularizing against class selectivity could yield three possible outcomes: If the previously-reported anti-correlation between selectivity and generalization is causal, then test accuracy should increase. But if class selectivity is necessary for high-fidelity class representations, then we should observe a decrease in test accuracy. Finally, if class selectivity is an emergent phenomenon and/or irrelevant to network performance, test accuracy should remain unchanged.

Surprisingly, we observed that reducing selectivity significantly improves test accuracy in ResNet18 trained on Tiny ImageNet for all examined values of $\alpha \in [-0.1, -2.5]$ (Figure 1; $p < 0.01$, Bonferroni-corrected t-test). Test accuracy increases with the magnitude of $\alpha$, reaching a maximum at $\alpha = -1.0$ (test accuracy at $\alpha_{-1.0} = 53.60 \pm 0.13$, $\alpha_0$ (i.e. no regularization) $= 51.57 \pm 0.18$), at which point there is a 1.6x reduction in class selectivity (mean class selectivity at $\alpha_{-1.0} = 0.22 \pm 0.0009$, $\alpha_0 = 0.35 \pm 0.0007$). Test accuracy then begins to decline; at $\alpha_{-3.0}$ test accuracy is statistically indistinct from $\alpha_0$, despite a 3x decrease in class class selectivity (mean class selectivity at $\alpha_{-3.0} = 0.12 \pm 0.0007$, $\alpha_0 = 0.35 \pm 0.0007$). Further reducing class selectivity beyond $\alpha = -3.5$ (mean class selectivity $= 0.10 \pm 0.0007$) has increasingly detrimental effects on test accuracy. These results show that the amount of class selectivity naturally learned by a network (i.e. the amount learned in the absence of explicit regularization) can actually constrain the network's performance.

ResNet20 trained on CIFAR10 also learned superfluous class selectivity. Although reducing class selectivity does not improve performance, it causes minimal detriment, except at extreme regularization scales ($\alpha \le -30$; Figure A2). Increasing the magnitude of $\alpha$ decreases mean class selectivity across the network, with little impact on test accuracy until mean class selectivity reaches $0.003 \pm 0.0002$ at $\alpha_{-30}$ (Figure A1d). Reducing class selectivity only begins to have a statistically significant effect on performance at $\alpha_{-1.0}$ (Figure A2a), at which point mean class selectivity across the network has decreased from $0.22 \pm 0.002$ at $\alpha_0$ (i.e. no regularization) to $0.07 \pm 0.0013$ at $\alpha_{-1.0}$—a factor of more than 3 (Figure A2c; $p = 0.03$, Bonferroni-corrected t-test). This implies that ResNet20 learns more than three times the amount of class selectivity required for maximum test accuracy.

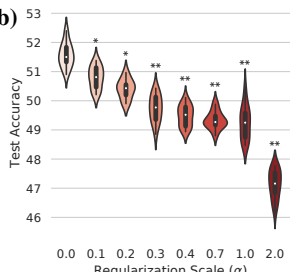 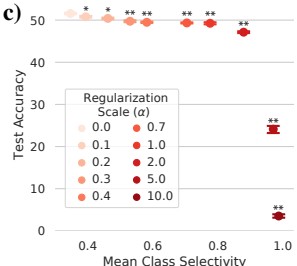

**Figure 3: Effects of increasing class selectivity on test accuracy in ResNet18 trained on Tiny ImageNet.** (**a**) Test accuracy (y-axis) as a function of regularization scale ($\alpha$; x-axis, intensity of red). (**b**) Identical to (**a**), but for a subset of $\alpha$ values. Each violin plot contains a boxplot in which the darker central lines denote the central two quartiles. (**c**) Test accuracy (y-axis) as a function of mean class selectivity (x-axis) across $\alpha$ values. Error bars denote 95% confidence intervals. $*p < 6 \times 10^{-5}$, $**p < 8 \times 10^{-12}$ difference from $\alpha = 0$, t-test, Bonferroni-corrected. See Appendix A.9 and A.12 for ResNet20 and VGG results, respectively.

We observed qualitatively similar results for VGG16 (see Appendix A.12). Although the difference is significant at $\alpha = -0.1$ ($p = 0.004$, Bonferroni-corrected t-test), it is possible to reduce mean class selectivity by a factor of 5 with only a 0.5% decrease in test accuracy, and by a factor of 10 with only a ~1% drop in test accuracy. These differences may be due to VGG16's naturally higher levels of class selectivity (see Figure A16 for comparisons between VGG16 and ResNet20). We also observed qualitatively similar results when using our regularization approach to decrease $d'$, a measure of class discriminability, in ResNet20 trained on CIFAR10 and ResNet18 trained on Tiny ImageNet (Appendix A.14). Furthermore, we also find that regularizing to reduce class selectivity improves test accuracy in ResNet50 trained on Tiny ImageNet (Appendix A.18). Together, these results demonstrate that class selectivity in individual units is largely unnecessary for optimal performance in CNNs trained on image classification tasks.

## 4.2 DOES SELECTIVITY SHIFT TO A DIFFERENT BASIS SET?

We were able to reduce mean class selectivity in all examined networks by a factor of at least three with minimal negative impact on test accuracy (~1%, at worst, for VGG16). However, one trivial solution for reducing class selectivity is for the network to "hide" it from the regularizer by rotating it off of unit-aligned axes or performing some other linear transformation. In this scenario the selectivity in individual units would be reduced, but remain accessible through linear combinations of activity across units. In order to test this possibility, we used CCA (see Appendix A.3), which is invariant to rotation and other invertible affine transformations, to compare the representations in regularized (i.e. low-selectivity) networks to the representations in unregularized networks.

We first established a meaningful baseline for comparison by computing the CCA distances between each pair of 20 replicate networks for a given value of $\alpha$ (we refer to this set of distances as $\rho(\alpha_r, \alpha_r)$). If regularizing against class selectivity causes the network to move selectivity off-axis, the CCA distances between regularized and unregularized networks —which we term $\rho(\alpha_r, \alpha_0)$—should be similar to $\rho(\alpha_r, \alpha_r)$. Alternatively, if class selectivity is suppressed via some non-affine transformation of the representation, $\rho(\alpha_r, \alpha_0)$ should exceed $\rho(\alpha_r, \alpha_r)$.

Our analyses confirm the latter hypothesis: we find that $\rho(\alpha_r, \alpha_0)$ significantly exceeds $\rho(\alpha_r, \alpha_r)$ for all values of $\alpha$ except $\alpha = -0.1$ in ResNet18 trained on Tiny ImageNet (Figure 2 $p < 1.3 \times 10^{-5}$, paired t-test). The effect is even more striking in ResNet20 trained on CIFAR10; all tested values of $\alpha$ are significant (Figure A3; $p < 5 \times 10^6$, paired t-test). Furthermore, the size of the effect is proportional to $\alpha$ in both models; larger $\alpha$ values yield representations that are more dissimilar to unregularized representations. These results support the conclusion that our regularizer doesn't just cause class selectivity to be rotated off of unit-aligned axes, but also suppresses it.

As an additional control to ensure that our regularizer did not simply shift class selectivity to off-axis directions in activation space, we calculated an upper bound on the amount of class selectivity that could be recovered by finding the linear projection of unit activations that maximizes class selectivity (see Appendix A.7 for methodological details). For both ResNet18 trained on Tiny ImageNet (Figure 2c) and ResNet20 trained on CIFAR10 (Figure A4b), the amount of class selectivity in the optimized projection decreases as a function of increasing $|\alpha|$, indicating that regularizing against class selectivity does not simply rotate the selectivity off-axis. Interestingly, the upper bound on class selectivity is very similar across regularization scales in the final two convolutional layers in both models (Figure A4a; A4c), indicating that immediate proximity to the logits (output) may mitigate

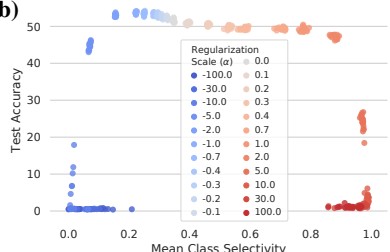

**Figure 4: Increasing class selectivity has deleterious effects on test accuracy compared to reducing class selectivity.** (a) Test accuracy (y-axis) as a function of regularization scale magnitude ($|\alpha|$) for negative (blue) vs positive (red) values of $\alpha$. Solid line in distributions denotes mean, dashed line denotes central two quartiles. **$p < 6 \times 10^{-6}$ difference between $\alpha < 0$ and $\alpha > 0$, Wilcoxon rank-sum test, Bonferroni-corrected. (b) Test accuracy (y-axis) as a function of mean class selectivity (x-axis). All results shown are for ResNet18.

the effect of class selectivity regularization. While we also found that the amount of class selectivity in the optimized projection is consistently higher than the observed axis-aligned class selectivity, we consider this to be an expected result, as the optimized projection represents an upper bound on the amount class selectivity that could be recovered from the models' representations. However, the decreasing upper bound as a function of increasing $|\alpha|$ indicates that our class selectivity regularizer decreases selectivity across all basis sets, and not just along unit-aligned axes.

### 4.3    INCREASED CLASS SELECTIVITY CONSIDERED HARMFUL

We have demonstrated that class selectivity can be significantly reduced with minimal impact on test accuracy. However, we only examined the effects of *reducing* selectivity. What are the effects of *increasing* selectivity? We examined this question by regularizing *for* class selectivity, instead of against it. This is achieved quite easily, as it requires only a change in the sign of $\alpha$. We first confirmed that changing the sign of the scale term in the loss function causes the intended effect of increasing class selectivity in individual units (see Appendix A.8).

Despite class selectivity not being strictly necessary for high performance, its ubiquity across biological and artificial neural networks leads us to suspect it may still be sufficient. We thus expect that increasing it would either improve test accuracy or yield no effect. For the same reason, we would consider it unexpected if increasing selectivity impairs test accuracy.

Surprisingly, we observe the latter outcome: increasing class selectivity negatively impacts network performance in ResNet18 trained on Tiny ImageNet (Figure 3a). Scaling the regularization has an immediate effect: a significant decline in test accuracy is present even at the smallest tested value of $\alpha$ ($p \leq 6 \times 10^{-5}$ for all $\alpha$, Bonferroni-corrected t-test) and falls catastrophically to $\sim$25% by $\alpha = 5.0$. The effect proceeds even more dramatically in ResNet20 trained on CIFAR10 (Figure A6a). Note that we observed a correlation between the strength of regularization and the presence of dead units in ResNet20 (but not ResNet18), however further analyses ruled this out as an explanation for the decline in test accuracy (see Appendix A.10).

One solution to generate a very high selectivity index is if a unit is silent for the vast majority of inputs and has low activations for remaining set of inputs. If this were the case, we would expect that regularizing to increase selectivity would cause units to be silent for the majority of inputs. However, we found that the majority of units were active for $\geq$80% of inputs even at $\alpha = 0.7$, after significant performance deficits have emerged in both ResNet18 and ResNet20 (Appendix A.11). These findings rule out sparsity as a potential explanation for our results. It is also possible that regularizing to increase class selectivity could discourage individual neurons from changing their preferred class during training, even if changing their preferred class would improve performance. If regularizing to increase class selectivity did indeed lock units in to their initial preferred class, this could impose a constraint on performance. We tested for this possibility in two ways (Appendix A.16): by examining the statistics of units' changes in preferred class during training (Figures A28, A29), and slowly warming up class selectivity regularization over the initial training epochs (Figures A30; A31). Approximately 100% of units across all examined models and regularization scales change their preferred class at least once during training (Figure A28), and the relationship between class selectivity regularization and the number of changes in a unit's preferred class over training is inconsistent (Figure A29). Furthermore, warming up the regularization has qualitatively similar effects as using a constant $\alpha$ (Figure A31). None of these analyses indicate that an inability to change preferred locations can fully explain the class selectivity-induced test accuracy impairment (see Appendix A.16 for additional details).

The effects of regularizing to increase class selectivity are qualitatively similar for VGG16 (see Appendix A.12) and ResNet50 (Appendix A.18); we observed across all models that increasing class selectivity beyond the levels that are learned naturally (i.e. without regularization, $\alpha = 0$) impairs network performance.

**Recapitulation**  We directly compare the effects of increasing vs. decreasing class selectivity in Figure 4 and Appendix A.17. The effects diverge immediately at $|\alpha| = 0.1$, and suppressing class selectivity yields a 6% increase in test accuracy relative to increasing class selectivity by $|\alpha| = 2.0$.

## 5  DISCUSSION

We examined the causal role of class selectivity in CNN performance by adding a term to the loss function that allows us to directly manipulate class selectivity across all neurons in the network. We found that class selectivity is not strictly necessary for networks to function, and that reducing it can even improve test accuracy. In ResNet18 trained on Tiny Imagenet, reducing class selectivity by $1.6\times$ improved test accuracy by over 2%. In ResNet20 trained on CIFAR10, we could reduce the mean class selectivity of units in a network by factor of $\sim2.5$ with no impact on test accuracy, and by a factor of $\sim20$—nearly to a mean of 0—with only a 2% change in test accuracy. We confirmed that our regularizer seems to suppress class selectivity, and not simply cause the network to rotate it off of unit-aligned axes. We also found that regularizing a network to increase class selectivity in individual units has negative effects on performance. These results resolve questions about class selectivity that remained inaccessible to previous approaches: class selectivity in individual units is neither necessary nor sufficient for—and can sometimes even constrain—CNN performance.

One caveat to our results is that they are limited to CNNs trained to perform image classification. It's possible that our findings are due to idiosyncrasies of benchmark datasets, and wouldn't generalize to more naturalistic datasets and tasks. Given that class selectivity is ubiquitous across DNNs trained on different tasks and datasets, future work should examine how broadly our results generalize, and the viability of class selectivity regularization as a general-purpose tool to improve DNN performance.

The presence of non-selective units in a network trained to perform image classification could appear puzzling. It may be difficult to envision how non-selective units could shape representations in a manner that is useful for a classification task. One possibility is that the class-conditional joint distribution of activations across units facilitates readout. Put another way, the correlations between units' activations can help separate the distributions of activations for different classes. Indeed, there is evidence that correlated variability between neurons can facilitate information readout in the brain (Zylberberg et al., 2016; Leavitt et al., 2017; Zylberberg, 2018; Nogueira et al., 2020).

We know that class selectivity in individual units naturally emerges over the course of learning. The single unit ablation studies show that class selectivity can have an effect on the performance of trained networks. And while it is possible that networks trained with selectivity regularization learn different solutions from networks trained without it, our results show that class selectivity is not strictly necessary for networks to learn representations that result in class-selective units. This finding naturally leads to a compelling question: if class selectivity is unnecessary, why does it emerge?

Our results also make a broader point about the potential pitfalls of focusing on the properties of single units when trying to understand DNNs, emphasizing instead the importance of analyses that focus on *distributed* representations. While we consider it essential to find tractable, intuitive approaches for understanding complex systems, it's critical to empirically verify that these approaches actually reflect functionally relevant properties of the system being examined.

ACKNOWLEDGEMENTS

We would like to thank Tatiana Likhomanenko, Tiffany Cai, Eric Mintun, Janice Lan, Mike Rabbat, Sergey Edunov, Yuandong Tian, and Lyndon Duong for their productive scrutiny and insightful feedback.

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

# A    APPENDIX

## A.1    MODELS, TRAINING, DATASETS, AND SOFTWARE

Our experiments were performed on ResNet18 and ResNet50 (He et al., 2016) trained on Tiny Imagenet (Fei-Fei et al., 2015), and ResNet20 (He et al. (2016); code modified from Idelbayev (2020)) and a VGG16-like network (Simonyan and Zisserman, 2015), both trained on CIFAR10 (Krizhevsky, 2009). All models were trained using stochastic gradient descent (SGD) with momentum = 0.9 and weight decay = 0.0001.

The maxpool layer after the first batchnorm layer (see He et al. (2016)) was removed because of the smaller size of Tiny Imagenet images compared to standard ImageNet images (64x64 vs. 256x256, respectively). ResNet18 were trained for 90 epochs with a minibatch size of 4096 samples with a learning rate of 0.1, multiplied (annealed) by 0.1 at epochs 35, 50, 65, and 80. ResNet50 was trained identically, except with a batch size of 1400 samples. Tiny Imagenet (Fei-Fei et al., 2015) consists of 500 training images and 50 images for each of its 200 classes. We used the validation set for testing and created a new validation set by taking 50 images per class from the training set, selected randomly for each training run.

The VGG16-like network is identical to the batch norm VGG16 in Simonyan and Zisserman (2015), except the final two fully-connected layers of 4096 units each were replaced with a single 512-unit layer. ResNet20 and VGG16 were trained for 200 epochs using a minibatch size of 256 samples. ResNet20 were trained with a learning rate of 0.1 and VGG16 with a learning rate of 0.01, both annealed by $10^{-1}$ at epochs 100 and 150. We split the 50k CIFAR10 training samples into a 45k sample training set and a 5k validation set, similar to our approach with Tiny Imagenet.

All experimental results were derived from the test set with the parameters from the epoch that achieved the highest validation set accuracy over the training epochs. 20 replicates with different random seeds were run for each hyperparameter set, except for ResNet50, which only used 5 replicates per hyperparameter set. Selectivity regularization was not applied to the output (logit) layer, nor was the output layer included any of our analyses.

Experiments were conducted using PyTorch (Paszke et al., 2019), analyzed using the SciPy ecosystem (Virtanen et al., 2019), and visualized using Seaborn (Waskom et al., 2017).

## A.2    EFFECT OF SELECTIVITY REGULARIZER ON TRAINING TIME

We quantified the number of training epochs required to reach 95% of maximum test accuracy ($t^{95}$). The $t^{95}$ without selectivity regularization ($t^{95}_{\alpha=0}$) for ResNet20 is 45±15 epochs (median ± IQR). $\alpha$ in [-2, 0.7] had overlapping IQRs with $\alpha = 0$. For ResNet18, $t^{95}_{\alpha=0} = 35 \pm 1$, while $t^{95}$ for $\alpha$ in [-2, 1] was as high as 51±1.5. Beyond these ranges, the $t^{95}$ exceeded $1.5 \times t^{95}_{\alpha=0}$ and/or was highly variable.

## A.3    CCA

### A.3.1    AN INTUITION

We used Canonical Correlation Analysis (CCA) to examine the effects of class selectivity regularization on hiden layer representations. CCA is a statistical method that takes two sets of multidimensional variates and finds the linear combinations of these variates that have maximum correlation with each other (Hotelling, 1936). Critically, CCA is invariant to rotation and other invertible affine transformations. CCA has been productively applied to analyze and compare representations in (and between) biological and neural networks (Sussillo et al., 2015; Smith et al., 2015; Raghu et al., 2017; Morcos et al., 2018a; Gallego et al., 2018).

We use projection-weighted CCA (PWCCA), a variant of CCA introduced in Morcos et al. (2018a) that has been shown to be more robust to noise than traditional CCA and other CCA variants (though for brevity we just use the term "CCA" in the main text). PWCCA generates a scalar value, $\rho$, that can be thought of as the distance or dissimilarity between the two sets of multidimensional variates, $L_1$ and $L_2$. For example, if $L_2 = L_1$, then $\rho_{L_1,L_2} = 0$. Now let $R$ be a rotation matrix. Because CCA is invariant to rotation and other invertible affine transformations, if $L_2 = RL_1$ (i.e. if $L_2$

is a rotation of $L_1$), then $\rho_{L_1,L_2} = 0$. In contrast, traditional similarity metrics such as Pearson's Correlation and cosine similarity would obtain different values if $L_2 = L_1$ compared to $L_2 = RL_1$. We use the PWCCA implementation available at https://github.com/google/svcca/, as provided in Morcos et al. (2018a).

### A.3.2   OUR APPLICATION

As an example for the analyses in our experiments, $L_1$ is the activation matrix for a layer in a network that was not regularized against class selectivity (i.e. $\alpha = 0$), and $L_2$ is the activation matrix for the same layer in a network that was structured and initialized identically, but subject to regularization against class selectivity (i.e. $\alpha < 0$). If regularizing against class selectivity causes the network's representations to be rotated (or to undergo to some other invertible affine transformation), then $\rho_{L_1,L_2} = 0$. In practice $\rho_{L_1,L_2} > 0$ due to differences in random seeds and/or other stochastic factors in the training process, so we can determine a threshold value $\epsilon$ and say $\rho_{L_1,L_2} \leq \epsilon$. If regularizing against class selectivity instead causes a non-affine transformation to the network's representations, then $\rho_{L_1,L_2} > \epsilon$.

In our experiments we empirically establish a distribution of $\epsilon$ values by computing the PWCCA distances between $\rho_{L_{2a}L_{2b}}$, where $L_{2a}$ and $L_{2b}$ are two networks from the set of 20 replicates for a given hyperparameter combination that differ only in their initial random seed values (and thus have the same $\alpha$). This gives $\binom{20}{2} = 190$ values of $\epsilon$. We then compute the PWCCA distance between each $\{L_1, L_2\}$ replicate pair, yielding a distribution of $20 \times 20 = 400$ values of $\rho_{L_1,L_2}$, which we compare to the distribution of $\epsilon$.

### A.3.3   FORMALLY

For the case of our analyses, let us start with a dataset $X$, which consists of $M$ data samples $\{x_1, ...x_M\}$. Using the notation from Raghu et al. (2017), the scalar output (activation) of a single neuron $i$ on layer $\iota$ in response to each data sample collectively form the vector

$$z_i^\iota = (z(x_i^\iota(x_1), ..., x_i^\iota(x_M))$$

We then collect the activation vector $z_i^l$ of every neuron in layer $\iota$ into a matrix $L = \{z_1^\iota, ..., z_M^\iota\}$ of size $N \times M$, $N$ is the number of neurons in layer $\iota$, and $M$ is the number of data samples. Given two such activation matrices $L_1$, of size $N_a \times M$, and $L_2$, of size $N_b \times M$, CCA finds the vectors $w$ (in $\mathbb{R}^{N_a}$) and $s$ (in $\mathbb{R}^{N_b}$), such that the inner product

$$\rho = 1 - \frac{\langle w^T L_1, s^T L_2 \rangle}{\|w^T L_1\| \cdot \|s^T L_2\|}$$

is maximized.

## A.4 REGULARIZING TO DECREASE CLASS SELECTIVITY IN RESNET18 AND RESNET20

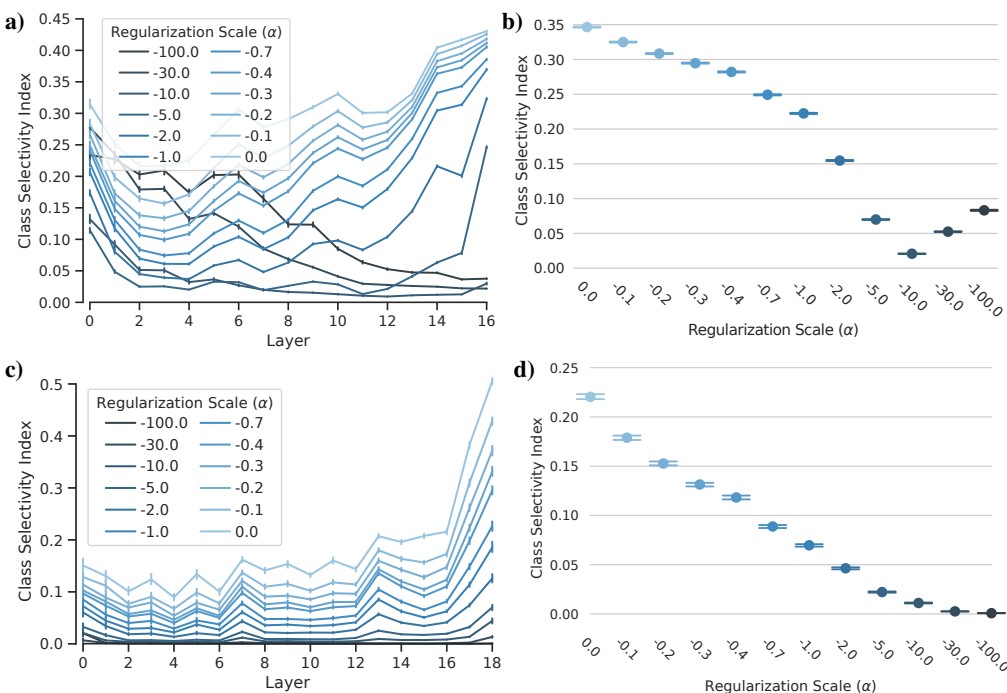

**Figure A1: Manipulating class selectivity by regularizing against it in the loss function.** (a) Mean class selectivity index (y-axis) as a function of layer (x-axis) for different regularization scales ($\alpha$; denoted by intensity of blue) for ResNet18. (b) Similar to (a), but mean is computed across all units in a network instead of per layer. (b) Similar to (a), but mean is computed across all units in a network instead of per layer. (c) and (d) are identical to (a) and (b), respectively, but for ResNet20. Error bars denote bootstrapped 95% confidence intervals.

## A.5 DECREASING CLASS SELECTIVITY WITHOUT DECREASING TEST ACCURACY IN RESNET20

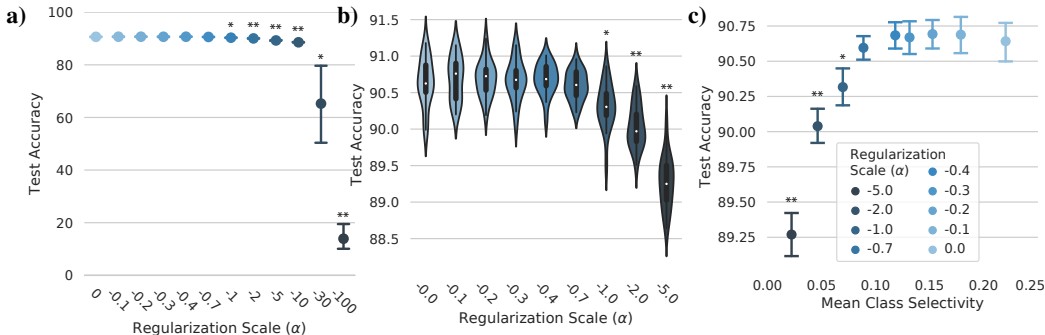

**Figure A2: Effects of reducing class selectivity on test accuracy in ResNet20 trained on CIFAR10.** (a) Test accuracy (y-axis) as a function of regularization scale ($\alpha$, x-axis and intensity of blue). (b) Identical to (a), but for a subset of $\alpha$ values. The center of each violin plot contains a boxplot, in which the darker central lines denote the central two quartiles. (c) Test accuracy (y-axis) as a function of mean class selectivity (x-axis) for different values of $\alpha$. Error bars denote 95% confidence intervals. $*p < 0.05$, $**p < 5 \times 10^{-6}$ difference from $\alpha = 0$, t-test, Bonferroni-corrected.

## A.6 CCA Results for ResNet20

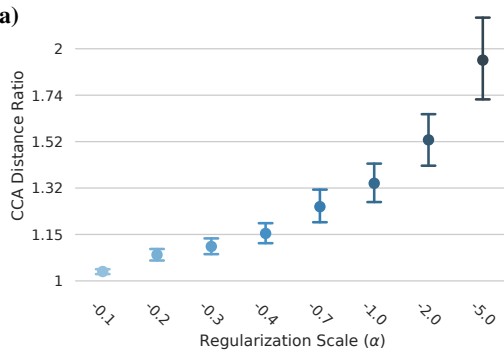

**Figure A3: Using CCA to check whether class selectivity is rotated off-axis in ResNet20 trained on CIFAR10.** Similar to Figure 2, we plot the average CCA distance ratio (y-axis) as a function of $\alpha$ (x-axis, intensity of blue). The distance ratio is significantly greater than the baseline for all values of $\alpha$ ($p < 5 \times 10^{-6}$, paired t-test). Error bars = 95% confidence intervals.

## A.7 Calculating an upper bound for off-axis selectivity

As an additional control to ensure that our regularizer did not simply shift class selectivity to off-axis directions in activation space, we calculated an upper bound on the amount of class selectivity that could be recovered by finding the linear projection of unit activations that maximizes class selectivity. To do so, we first collected the validation set activation vector $z_i^l$ of every neuron in layer $\iota$ into a matrix $A_{val} = \{z_1^\iota, ..., z_M^\iota\}$ of size $M \times N$, where $M$ is the number of data samples in validation set and $N$ is the number of neurons in layer $\iota$. We then found the projection matrix $W \in \mathbb{R}^{N \times N}$ that minimizes the loss

$$loss = (1 - SI(A_{val}W))$$

such that

$$||W^T W - I||^2 = 0$$

i.e. $W$ is orthonormal, where $SI$ is the selectivity index from Equation 1. We constrained $W$ to be orthonormal because the non-orthonormal solution to maximizing selectivity is degenerate: project all axes onto the single direction in activation space with the highest class selectivity. We used Lezcano-Casado (2019)'s toolbox to constrain $W$ to be orthonormal. Because $SI$ requires inputs $\geq 0$, we shifted the columns of $AW$ by subtracting the columnwise mininum value before computing $SI$. The optimization was performed using Adam (Kingma and Ba, 2015) with a learning rate of 0.001 for 3500 steps or until the magnitude of the change in loss was less than $10^{-6}$ for 10 steps. $W$ was then used to project the activation matrix for the test set $A_{test}$, and the selectivity index was calculated for each axis of the new activation space (i.e. each column of $A_{test}W$) after shifting the columns of $A_{test}W$ to be $\geq 0$. A separate $W$ was obtained for each layer of each model and for each replicate and value of $\alpha$.

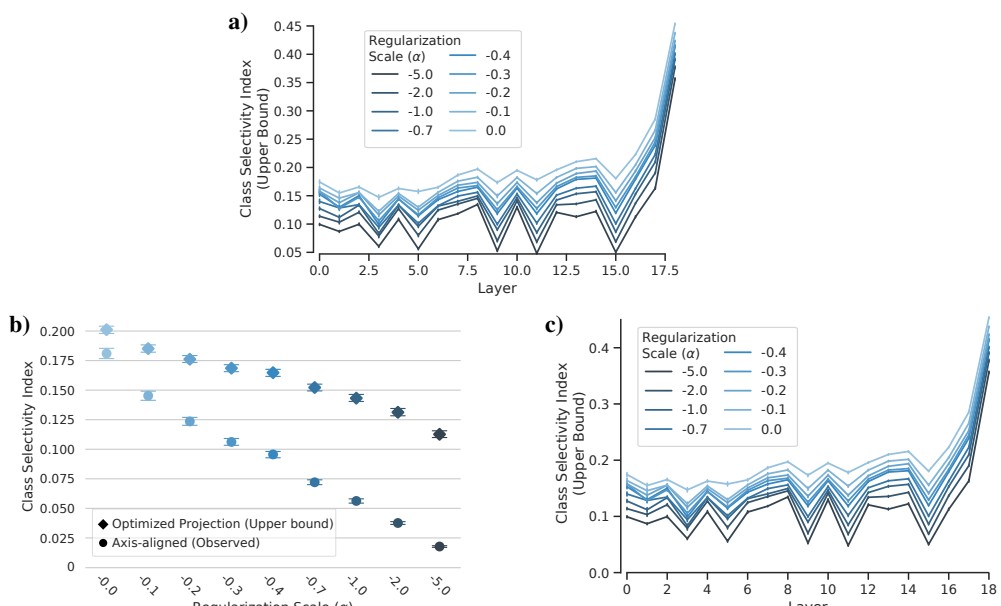

**Figure A4: An upper bound for off-axis class selectivity.** **(a)** Upper bound on class selectivity (y-axis) as a function of layer (x-axis) for different regularization scales ($\alpha$; denoted by intensity of blue) for ResNet18 trained on Tiny ImageNet. **(b)** Mean class selectivity (y-axis) as a function of regularization scale ($\alpha$; x-axis) for ResNet20 trained on CIFAR10. Diamond-shaped data points denote the upper bound on class selectivity for a linear projection of activations as described in Appendix A.7, while circular points denote the amount of axis-aligned class selectivity for the corresponding values of $\alpha$. **(c) (a)**, but for ResNet20 trained on CIFAR10. Error bars = 95% confidence intervals.

## A.8   REGULARIZING TO INCREASE CLASS SELECTIVITY IN RESNET18 AND RESNET20

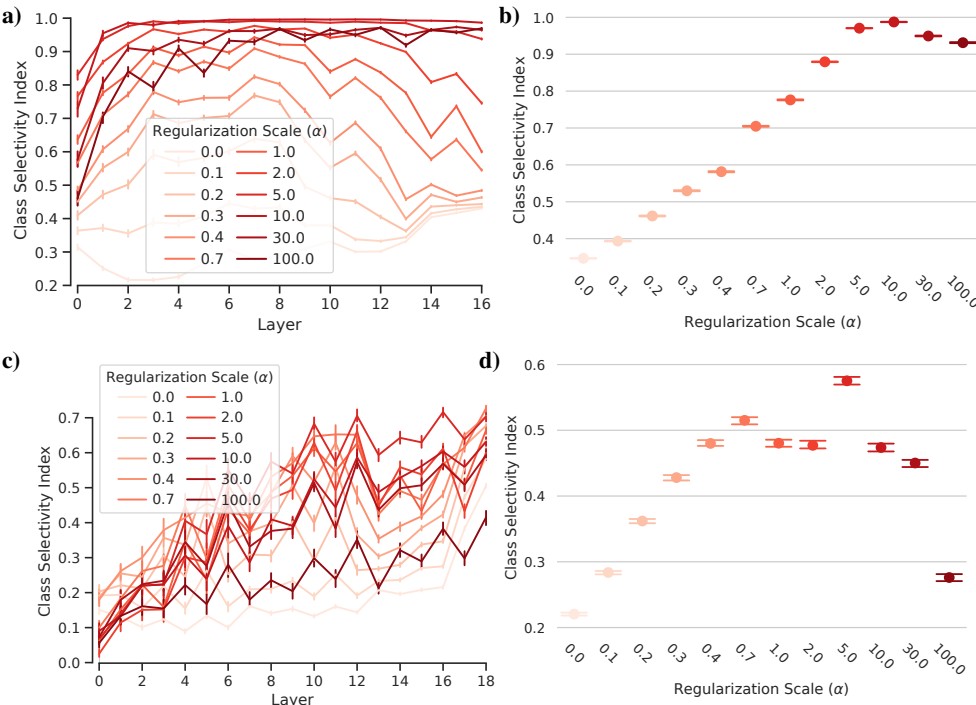

**Figure A5: Regularizing to increase class selectivity** (a) Mean class selectivity index (y-axis) as a function of layer (x-axis) for different regularization scales ($\alpha$; denoted by intensity of red) for ResNet18. (b) Similar to (a), but mean is computed across all units in a network instead of per layer. (c) and (d) are identical to (a) and (b), respectively, but for ResNet20. Note that the inconsistent effect of larger $\alpha$ values in (c) and (d) is addressed in Appendix A.10. Error bars denote bootstrapped 95% confidence intervals.

## A.9 ADDITIONAL EFFECTS OF CLASS SELECTIVITY REGULARIZATION ON TEST ACCURACY IN RESNET20

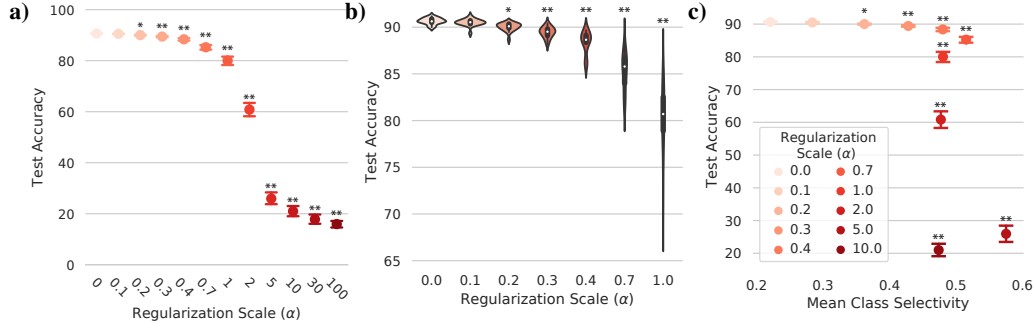

**Figure A6: Effects of increasing class selectivity on test accuracy on ResNet20 trained on CIFAR10. (a)** Test accuracy (y-axis) as a function of regularization scale ($\alpha$; x-axis, intensity of red). **(b)** Identical to **(a)**, but for a subset of $\alpha$ values. The center of each violin plot contains a boxplot, in which the darker central lines denote the central two quartiles. **(c)** Test accuracy (y-axis) as a function of mean class selectivity (x-axis) for different values of $\alpha$. Error bars denote 95% confidence intervals. $*p < 2 \times 10^{-4}$, $**p < 5 \times 10^{-7}$ difference from $\alpha = 0$, t-test, Bonferroni-corrected.

## A.10 SINGLE UNIT NECROMANCY

**Lethal ReLUs**  The inconsistent relationship between $\alpha$ and class selectivity for larger values of $\alpha$ led us to question whether the performance deficits were due to an alternative factor, such as the optimization process, rather than class selectivity per se. Interestingly, we observed that ResNet20 regularized to increase selectivity contained significantly higher proportions of dead units, and the number of dead units is roughly proportional to alpha (see Figure A8a). Regularizing to increase class selectivity did not cause units to die in ResNet18 trained on Tiny ImageNet except at very extreme values of $\alpha$ ($\alpha \geq 30$), though even at these values the proportion of dead units never exceeded 0.03 (Figure A7). Removing the dead units in ResNet20 makes the relationship between regularization and selectivity in ResNet20 more consistent at large regularization scales (see Appendix A8).

The presence of dead units is not unexpected, as units with the ReLU activation function are known to suffer from the "dying ReLU problem"(Lu et al., 2019): If, during training, a weight update causes a unit to cease activating in response to all training samples, the unit will be unaffected by subsequent weight updates because the ReLU gradient at $x \leq 0$ is zero, and thus the unit's activation will forever remain zero. The dead units could explain the decrease in performance from regularizing to increase selectivity as simply a decrease in model capacity.

**Fruitless resuscitation**  One solution to the dying ReLU problem is to use a leaky-ReLU activation function (Maas et al., 2013), which has a non-zero slope, $b$ (and thus non-zero gradient) for $x \leq 0$. Accordingly, we re-ran the previous experiment using units with a leaky-ReLU activation in an attempt to control for the potential confound of dead units. Note that because the class selectivity index assumes activations $\geq 0$, we shifted activations by subtracting the minimum activation when computing selectivity for leaky-ReLUs. If the performance deficits from regularizing for selectivity are simply due to dead units, then using leaky-ReLUs should rescue performance. Alternatively, if dead units are not the cause of the performance deficits, then leaky-ReLUs should not have an effect.

We first confirmed that using leaky-ReLUs solves the dead unit problem. Indeed, the proportion of dead units is reduced to 0 in all networks across all tested values of $b$. Despite complete recovery of the dead units, however, using leaky-ReLUs does not rescue class selectivity-induced performance deficits (Figure A9). While the largest negative slope value improved test accuracy for larger values of $\alpha$, the improvement was minor, and increasing $\alpha$ still had catastrophic effects. These results confirm that dead units cannot explain the rapid and catastrophic effects of increased class selectivity on performance.

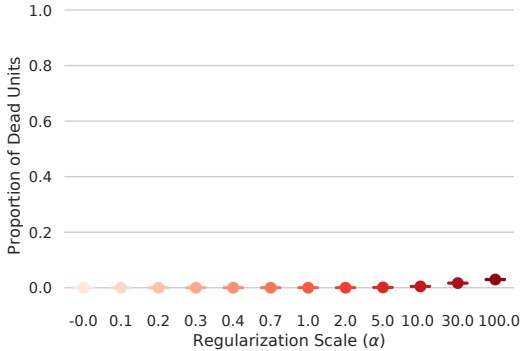

**Figure A7: Regularizing to increase class selectivity does not cause units to die in ResNet18 trained on Tiny ImageNet**. (**a**) Proportion of dead units (y-axis) for different for different regularization scales ($\alpha$; x-axis, intensity of red). Error bars denote 95% confidence intervals.

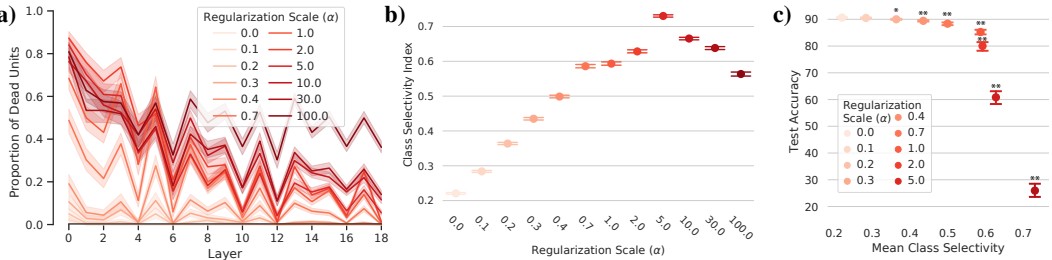

**Figure A8: Removing dead units partially stabilizes the effects of large positive regularization scales in ResNet20.** (**a**) Proportion of dead units (y-axis) as a function of layer (x-axis) for different regularization scales ($\alpha$, intensity of red). (**b**) Mean class selectivity index (y-axis) as a function of regularization scale ($\alpha$; x-axis and intensity of red) after removing dead units. Removing dead units from the class selectivity calculation establishes a more consistent relationship between $\alpha$ and the mean class selectivity index (compare to Figure A5d). (**c**) Test accuracy (y-axis) as a function of mean class selectivity (x-axis) for different values of $\alpha$ after removing dead units from the class selectivity calculation. Error bars denote 95% confidence intervals. *$p < 2 \times 10^{-4}$, **$p < 5 \times 10^{-7}$ difference from $\alpha = 0$ difference from $\alpha = 0$, t-test, Bonferroni-corrected. All results shown are for ResNet20.

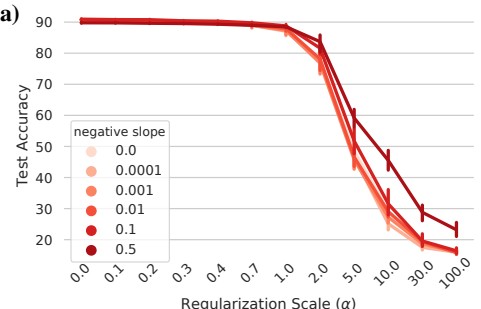 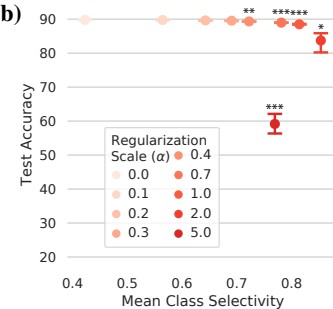

**Figure A9: Reviving dead units does not rescue the performance deficits caused by increasing selectivity in ResNet20.** (a) Test accuracy (y-axis) as a function of regularization scale ($\alpha$; x-axis) for different leaky-ReLU negative slopes (intensity of red). Leaky-ReLUs completely solve the dead unit problem but do not fully rescue test accuracy for networks with $\alpha > 0$. (b) Mean class selectivity index (y-axis) as a function of regularization scale ($\alpha$; x-axis and intensity of red) for leaky-ReLU negative slope = 0.5. *$p < 0.001$, **$p < 2 \times 10^{-4}$, ***$p < 5 \times 10^{-10}$ difference from $\alpha = 0$, t-test, Bonferroni-corrected. Error bars denote bootstrapped 95% confidence intervals.

## A.11 Ruling out a degenerate solution for increasing selectivity

One degenerate solution to generate a very high selectivity index is for a unit to be silent for the vast majority of inputs, and have low activations for the small set of remaining inputs. We refer to this scenario as "activation minimization". We verified that activation minimization does not fully account for our results by examining the proportion of activations which elicit non-zero activations in the units in our models. If our regularizer is indeed using activation minimization to generate high selectivity in individual units, then regularizing to increase class selectivity should cause most units to have non-zero activations for only a very small proportion of samples. In ResNet18 trained on Tiny ImageNet, we found that sparse units, defined as units that do not respond to at least half of the data samples, only constitute more than 10% of the total population at extreme positive regularization scales ($\alpha \geq 10$; Figure A10a), well after large performance deficits >4% emerge (Figure 3). In ResNet20 trained on CIFAR10, networks regularized to have higher class selectivity (i.e. positive $\alpha$) did indeed have more sparse units (Figure A10b). However, this effect does not explain away our findings: by $\alpha = 0.7$, the majority of units respond to over 80% of samples (i.e. they are not sparse), but test accuracy has already decreased by 5% (Figure A6). These results indicate that activation minimization does not explain class selectivity-related changes in test accuracy.

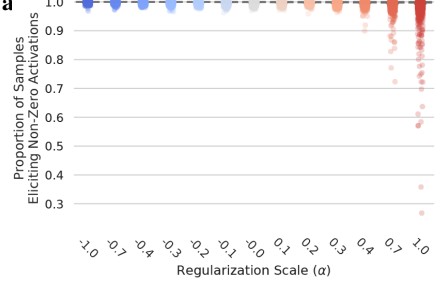 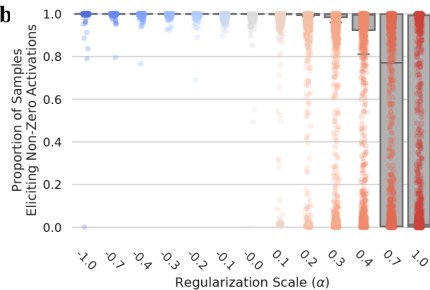

**Figure A10: Activation minimization does not explain selectivity-induced performance changes.** (a) Proportion of samples eliciting a non-zero activation (y-axis) vs. regularization scale ($\alpha$; x-axis) in ResNet18 trained on Tiny ImageNet. Data points denote individual units. Boxes denote IQR, whiskers extend 2×IQR. Note that the boxes are very compressed because the distribution is confined almost entirely to y=1.0 for all values of x. (b) Identical to (a), but for ResNet20 trained on CIFAR10.

## A.12 RESULTS FOR VGG16

Modulating class selectivity in VGG16 yielded results qualitatively similar to those we observed in ResNet20. The regularizer reliably decreases class selectivity for negative values of $\alpha$ (Figure A11), and class selectivity can be drastically reduced with little impact on test accuracy (Figure A12. Although test accuracy decreases significantly at $\alpha = -0.1$ ($p = 0.004$, Bonferroni-corrected t-test), the effect is small: it is possible to reduce mean class selectivity by a factor of 5 with only a 0.5% decrease in test accuracy, and by a factor of 10—to 0.03—with only a ∼1% drop in test accuracy.

Regularizing to increase class selectivity also has similar effects in VGG16 and ResNet20. Increasing $\alpha$ causes class selectivity to increase, and the effect becomes less consistent at large values of $\alpha$ (Figure A5). Although the class selectivity-induced collapse in test accuracy does not emerge quite as rapidly in VGG16 as it does in ResNet20, the decrease in test accuracy is still significant at the smallest tested value of $\alpha$ ($\alpha = 0.1$, $p = 0.02$, Bonferroni-corrected t-test), and the effects on test accuracy of regularizing to promote vs. discourage class selectivity become significantly different at $\alpha = 0.3$ ($p = 10^{-4}$, Wilcoxon rank-sum test; Figure A15). Our observations that class selectivity is neither necessary nor sufficient for performance in both VGG16 and ResNet20 indicates that this is likely a general property of CNNs.

It is worth noting that VGG16 exhibits greater class selectivity than ResNet20. In the absence of regularization (i.e. $\alpha = 0$), mean class selectivity in ResNet20 is 0.22, while in VGG16 it is 0.35, a 1.6x increase. This could explain why positive values of $\alpha$ seem to have a stronger effect on class selectivity in VGG16 relative to ResNet20 (compare Figure A5 and Figure A13; also see Figure A16b).

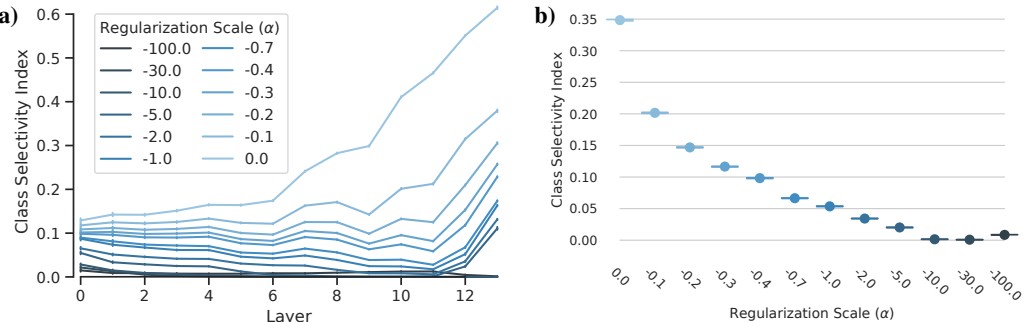

**Figure A11: Regularizing to decrease class selectivity in VGG16.** (**a**) Mean class selectivity index (y-axis) as a function of layer (x-axis) for different regularization scales ($\alpha$; denoted by intensity of blue) for VGG16. (**b**) Similar to (**a**), but mean is computed across all units in a network instead of per layer. Error bars denote bootstrapped 95% confidence intervals.

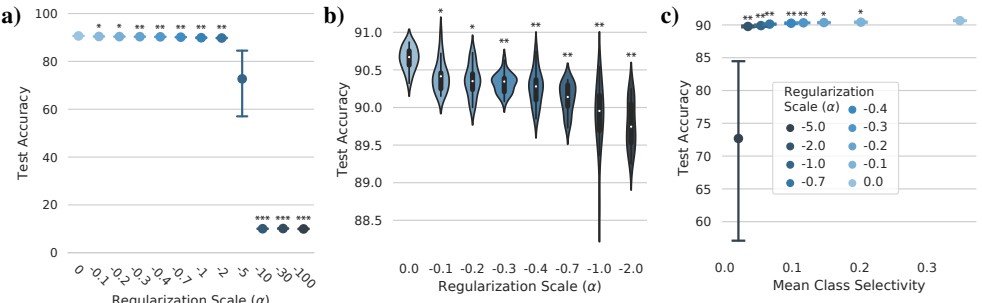

**Figure A12: Effects of reducing class selectivity on test accuracy in VGG16.** (**a**) Test accuracy (y-axis) as a function of regularization scale ($\alpha$, x-axis and intensity of blue). (**b**) Identical to (**a**), but for a subset of $\alpha$ values. The center of each violin plot contains a boxplot, in which the darker central lines denote the central two quartiles. (**c**) Test accuracy (y-axis) as a function of mean class selectivity (x-axis) for different values of $\alpha$. Error bars denote 95% confidence intervals. *$p < 0.005$, **$p < 5 \times 10^{-6}$, ***$p < 5 \times 10^{-60}$ difference from $\alpha = 0$, t-test, Bonferroni-corrected. All results shown are for VGG16.

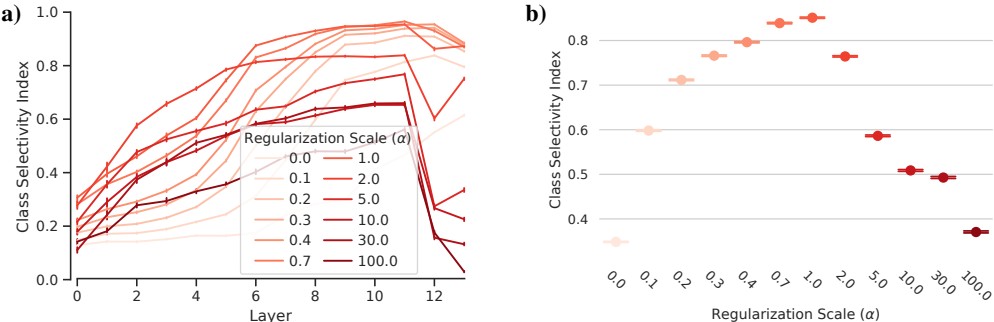

**Figure A13: Regularizing to increase class selectivity in VGG16.** (a) Mean class selectivity index (y-axis) as a function of layer (x-axis) for different regularization scales ($\alpha$; denoted by intensity of red) for VGG16. (b) Similar to (a), but mean is computed across all units in a network instead of per layer. Error bars denote bootstrapped 95% confidence intervals.

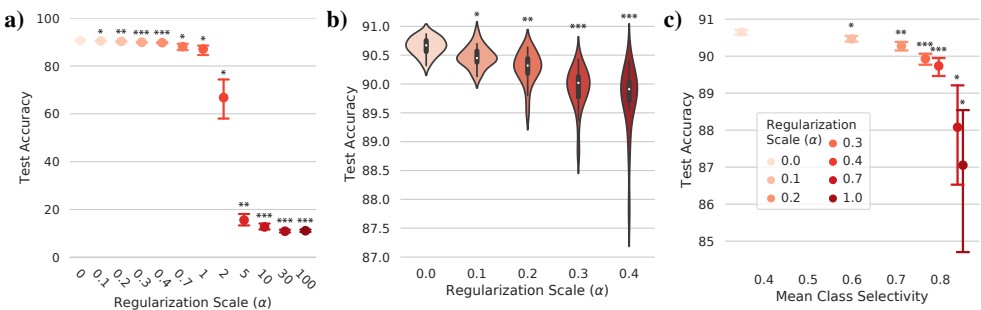

**Figure A14: Effects of increasing class selectivity on test accuracy in VGG16.** (a) Test accuracy (y-axis) as a function of regularization scale ($\alpha$, x-axis and intensity of red). (b) Identical to (a), but for a subset of $\alpha$ values. The center of each violin plot contains a boxplot, in which the darker central lines denote the central two quartiles. (c) Test accuracy (y-axis) as a function of mean class selectivity (x-axis) for different values of $\alpha$. Error bars denote 95% confidence intervals. $*p < 0.05$, $**p < 5 \times 10^{-4}$, $***p < 9 \times 10^{-6}$ difference from $\alpha = 0$, t-test, Bonferroni-corrected. All results shown are for VGG16.

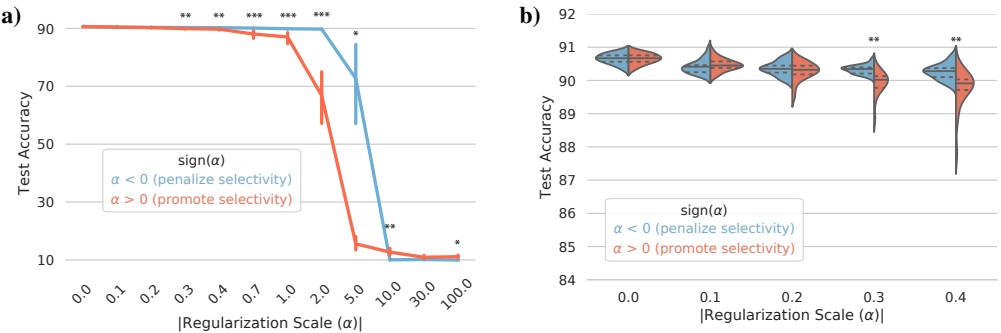

**Figure A15: Regularizing to promote vs. penalize class selectivity in VGG16.** (a) Test accuracy (y-axis) as a function of regularization scale magnitude ($|\alpha|$; x-axis) when promoting ($\alpha > 0$, red) or penalizing ($\alpha < 0$, blue) class selectivity in VGG16. Error bars denote bootstrapped 95% confidence intervals. (b) Identical to (a), but for a subset of $|\alpha|$ values. $*p < 0.05$, $**p < 10^{-3}$, $***p < 10^{-6}$ difference between $\alpha < 0$ and $\alpha > 0$, Wilcoxon rank-sum test, Bonferroni-corrected.

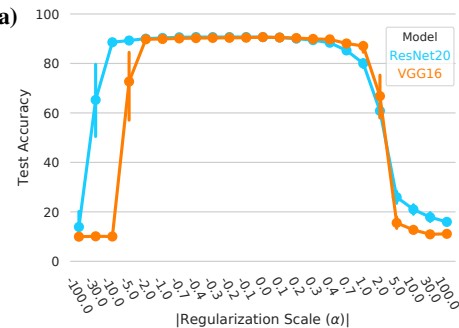 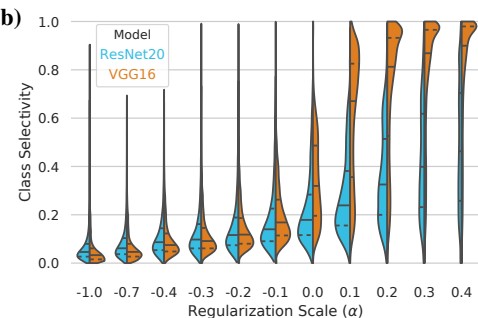

**Figure A16: Effects of class selectivity regularization in ResNet20 vs VGG16.** (**a**) Test accuracy (y-axis) as a function of regularization scale ($\alpha$ x-axis) for ResNet20 (cyan) and VGG16 (orange). Error bars denote bootstrapped 95% confidence intervals. (**b**) Class selectivity (y-axis) as a function of regularization scale ($\alpha$ for Resnet20 (cyan) and VGG16 (orange).

### A.13 ADDITIONAL MEASURES OF CLASS INFORMATION IN SINGLE UNITS

In order to confirm that the effect of the regularizer is not unique to our chosen class selectivity metric, we also examined the effect of our regularizer on two different measures of class information in single units: the "precision" metric for class selectivity (Zhou et al., 2015; 2018; Gale et al., 2019), and mutual information (Cover, 1999).

The precision metric is calculated by finding the $N$ images that most strongly activate a given unit, then finding the image class $C_i$ that constitutes the largest proportion of the $N$ images. Precision is defined as this proportion. For example, if $N = 200$, and the "cats" class, with 74 samples, constitutes the largest proportion of those 200 activations for a unit, then the precision of the unit is $\frac{74}{200} = 0.34$. Note that for a given number of classes $C$, precision is bounded by $[\frac{1}{C}, 1]$, thus in our experiments the lower bound on precision is 0.1. Zhou et al. (2015) used $N = 60$, while Gale et al. (2019) used $N = 100$. We chose to use the number of samples per class in the test set data and thus the largest possible sample size. This yielded $N = 1000$ for CIFAR10 and $N = 50$ for Tiny Imagenet.

The class selectivity regularizer has similar effects on precision as it does on the class selectivity index. Regularizing against class selectivity has a consistent effect on precision (Figure A17), while regularizing to promote class selectivity has a consistent effect in ResNet18 trained on Tiny ImageNet and for smaller values of $\alpha$ in ResNet20 trained on CIFAR10. However, the relationship between precision and the class selectivity index becomes less consistent for larger positive values of $\alpha$ in ResNet20 trained on CIFAR10. One explanation for this is that activation sparsity is a valid solution for maximizing the class selectivity index but not precision. For example, a unit that responded only to ten samples from the class "cat" and not at all to the remaining samples would have a class selectivity index of 1, but a precision value of 0.11. This seems likely given the increase in sparsity observed for very large positive values of $\alpha$ (see Appendix A.11).

The effects of the class selectivity regularizer on mutual information are nearly identical to its effects on precision. Regularizing for and against class selectivity has a consistent effect on mutual information in ResNet18 trained on Tiny ImageNet (Figure A18). Regularizing to reduce class selectivity also has a consistent effect in ResNet20 trained on CIFAR10, but the relationship between mutual information and the class selectivity index becomes less consistent for larger positive values of $\alpha$. However, given that class selectivity and mutual information are very different quantities, and that Morcos et al. (2018b) did not observe a clear relationship between mutual information, class selectivity, and individual unit importance (as measured by impact of ablation), we hesitate to make strong conclusions about the role of mutual information in our selectivity regularizer.

While there are additional class selectivity metrics that we could have used to further assess the effect of our regularizer, many of them are based on relating the activity of a neuron to the accuracy of the network's output(s) (e.g. top class selectivity Gale et al. (2019) and class correlation Li et al. (2015); Zhou et al. (2018)), confounding classification accuracy and class selectivity. Accordingly, these metrics are unfit for use in experiments that examine the relationship between class selectivity and classification accuracy, which is exactly what we do here.

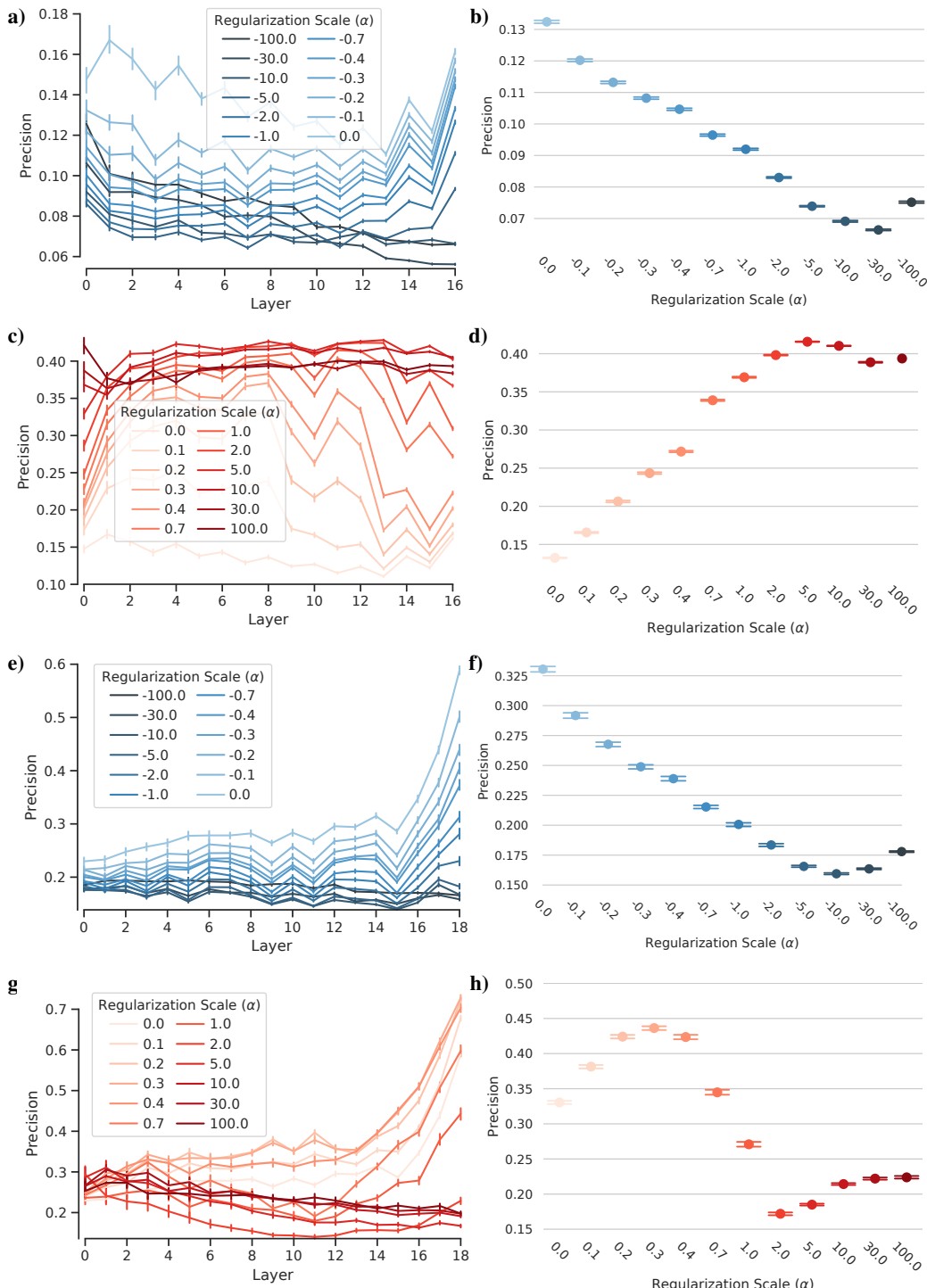

**Figure A17: Class selectivity regularization has similar effects when measured using a different class selectivity metric.** (a) Mean precision (y-axis) as a function of layer (x-axis) for different regularization scales ($\alpha$; denoted by intensity of blue) when regularizing against class selectivity in ResNet18. Precision is an alternative class selectivity metric (see Appendix A.13). (b) Similar to (a), but mean is computed across all units in a network instead of per layer. (c) and (d) are identical to (a) and (b), respectively, but when regularizing to promote class selectivity. (e-h) are identical to (a-d), respectively, but for ResNet20. Error bars denote bootstrapped 95% confidence intervals.

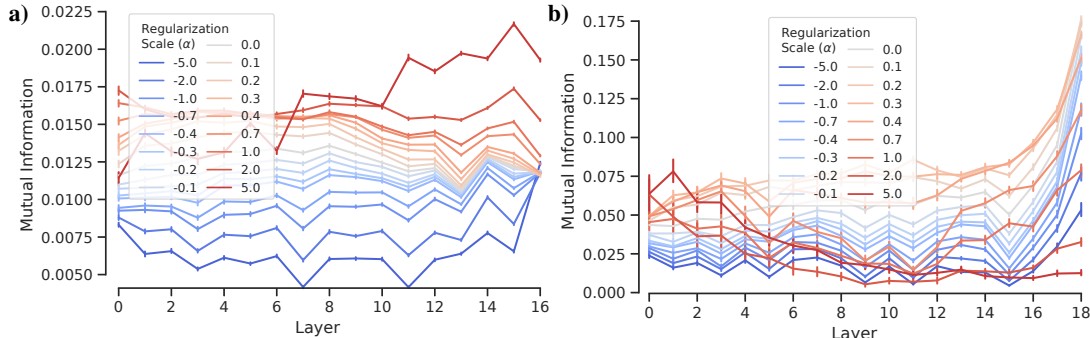

**Figure A18: Effects of class selectivity regularization on mutual information.** (**a**) Mean mutual information (y-axis) as a function of layer (x-axis) for different regularization scales ($\alpha$; negative values denoted by intensity of blue, positive values denoted by intensity of red) when regularizing to control class selectivity in ResNet18. (**b**) Identical to (**a**), but for ResNet20. Error bars denote bootstrapped 95% confidence intervals.

## A.14 REGULARIZING TO CONTROL CLASS DISCRIMINABILITY

We further confirmed that our results are not unique to our chosen class selectivity metric by examining a related, but distinct metric: class discriminability. $d'$ is a measure of the discriminability between two distributions (Stanislaw and Todorov, 1999; Macmillan and Creelman, 2004), defined as

$$d' = \frac{\mu_{max} - \mu_{-max}}{\sqrt{\frac{1}{2}(\sigma_{max}^2 + \sigma_{-max}^2) + \epsilon}}. \tag{4}$$

The numerator of $d'$ is identical to the numerator of the selectivity index (Equation 1)—the sum of $\mu_{max}$, the largest class-conditional mean activation, and $\mu_{-max}$, the mean response to the remaining (i.e. non-$\mu_{max}$) classes—but the denominator differs. $\sigma^2$ denotes the variance across activations.[1] $\epsilon$ is a small value to prevent division by zero (we used $10^{-7}$).

Because $d'$ is differentiable with respect to the model parameters, we can control the amount of class discriminability learned by individual units using the same approach as when regularizing to control class selectivity. Instead of calculating the class selectivity index for each unit, we calculated $d'$. This leads to the following loss function:

$$loss = -\sum_c^C y_c \cdot \log(\hat{y}_c) - \alpha \mu_{d'} \tag{5}$$

This loss is identical to Equation 2, except the selectivity term, $\mu_{SI}$, is replaced with $\mu_{d'}$:

$$\mu_{d'} = \frac{1}{L} \sum_l^L \frac{1}{U} \sum_u^U d'_u \tag{6}$$

As when computing $\mu_{SI}$ (Equation 3), $l$ is a convolutional layer, $L$ is the number of layers, $u$ is a unit (i.e. feature map), $U$ is the number of units in a given layer, and $u$ is a unit. The procedure for regularizing $d'$ is otherwise identical to regularizing class selectivity (Approach 3.3).

Regularizing to control $d'$ has the intended effect in both ResNet18 trained on Tiny ImageNet (Figure A19a) and ResNet20 trained on CIFAR10 (Figure A19c); the amount of class discriminability learned by individual units varies as a function of the sign and scale of regularization ($\alpha$). $d'$ also correlates strongly with class selectivity across units (Spearman's $\rho = 0.83$ for ResNet18 trained on Tiny ImageNet; $\rho = 0.90$ for ResNet20 trained on CIFAR10; $p < 10^{-5}$ for both). We observe very similar effects on test accuracy as when manipulating class selectivity: increasing $d'$ has rapid negative effects on test accuracy, while decreasing $d'$ has more modest effects. Though we do not observe an improvement in test accuracy as we did when regularizing to decrease class selectivity in ResNet18 trained on Tiny ImageNet, the asymmetry in effect between increasing vs. decreasing $d'$ is nevertheless consistent, indicating that neither class selectivity nor discriminability are sufficient nor strictly necessary for CNN performance.

---

[1]Traditional signal detection terminology describes $d'$ as being computed for a "signal" distribution ($\mu_{signal}, \sigma_{signal}$) and a "noise" distribution ($\mu_{noise}, \sigma_{noise}$). In our setting, $max$ is the signal distribution and $-max$ is the noise distribution.

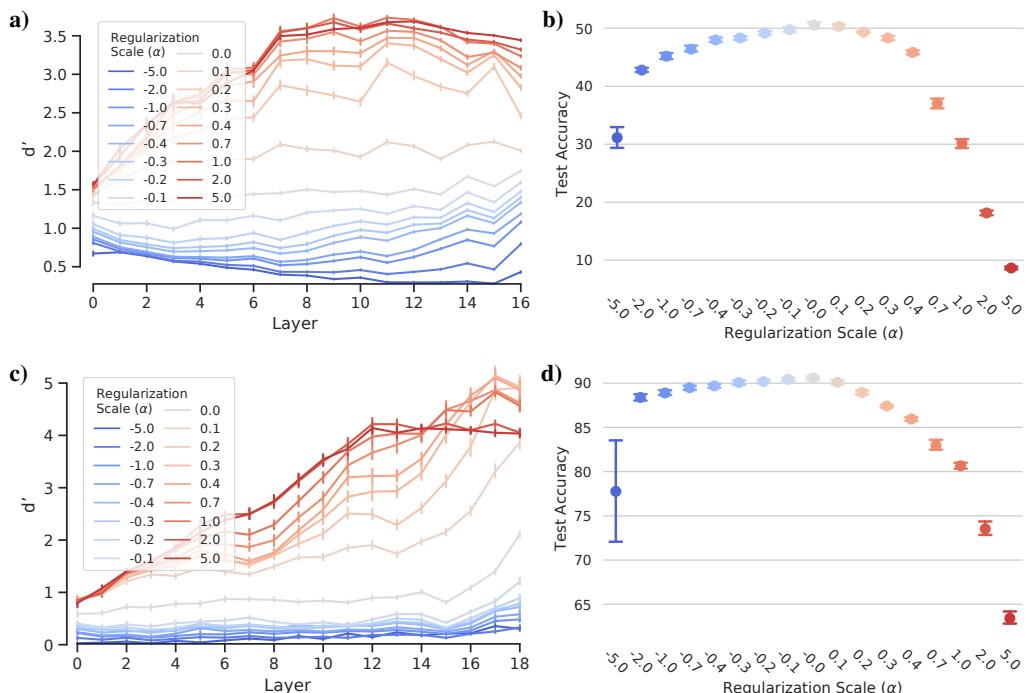

**Figure A19: Regularizing to control class discriminability ($d'$) in individual units.** (**a**) Mean class selectivity (y-axis) as a function of layer (x-axis) for different values of $\alpha$ (intensity of blue) when class selectivity regularization is restricted to the first three network layers in ResNet18 trained on Tiny ImageNet. (**b**) Mean class selectivity in the first three layers (y-axis) as a function of $\alpha$ (x-axis) in ResNet18 trained on Tiny ImageNet. (**c**) and (**d**) are identical to (**a**) and (**b**), respectively, but for ResNet20 trained on CIFAR10. Error bars = 95% confidence intervals.

## A.15 RESTRICTING CLASS SELECTIVITY REGULARIZATION TO THE FIRST THREE OR FINAL THREE LAYERS

To investigate the layer-specificity of the effects of class selectivity regularization, we also examined the effects of restricting class selectivity regularization to the first three or last three layers of the networks. Interestingly, we found that much of the effect of regularizing for or against selectivity on test accuracy was replicated even when the regularization was restricted to the first or final three layers. For example, reducing class selectivity in the first three layers either improves test accuracy—in ResNet18 trained on Tiny ImageNet—or has little-to-no effect on test accuracy—in ResNet20 trained on CIFAR10 (Figures A20 and A21). Likewise, regularizing to increase class selectivity in the first three layers had an immediate negative impact on test accuracy in both models (Figures A22 and A23). Regularizing against class selectivity in the final three layers (Figures A24 and A25) caused a modest increase in test accuracy over a narrow range of $\alpha$ in ResNet18 trained on Tiny ImageNet: less than half a percent gain at most (at $\alpha = -0.2$), and no longer present by $\alpha = -0.4$ (Figure A25b). In ResNet20, regularizing against class selectivity in the final three layers actually causes a decrease in test accuracy (Figures A21c and A21d). Given that the logit (output) layer of CNNs trained for image classification are by definition class-selective, we thought that regularizing to increase class selectivity in the final three layers could improve performance, but surprisingly it causes an immediate drop in test accuracy in both models (Figures A26 and A27). Our observation that regularizing to decrease class selectivity provides greater benefits (in the case of ResNet18) or less impairment (in the case of ResNet20) in the first three layers compared to the final three layers leads to the conclusion that class selectivity is less necessary (or more detrimental) in early layers compared to late layers.

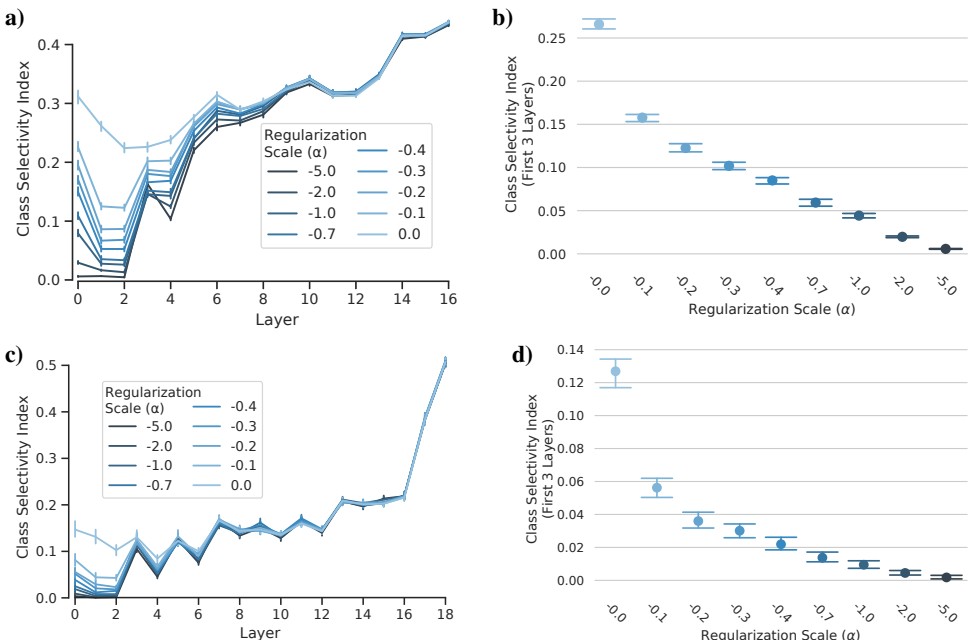

**Figure A20: Regularizing to decrease class selectivity in the first three network layers.** (a) Mean class selectivity (y-axis) as a function of layer (x-axis) for different values of $\alpha$ (intensity of blue) when class selectivity regularization is restricted to the first three network layers in ResNet18 trained on Tiny ImageNet. (b) Mean class selectivity in the first three layers (y-axis) as a function of $\alpha$ (x-axis) in ResNet18 trained on Tiny ImageNet. (c) and (d) are identical to (a) and (b), respectively, but for ResNet20 trained on CIFAR10. Error bars = 95% confidence intervals.

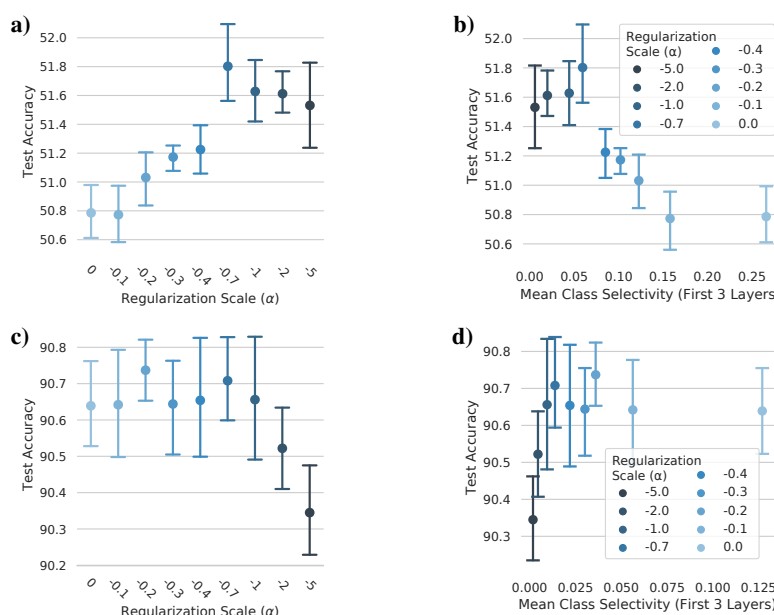

**Figure A21: Effects on test accuracy when regularizing to decrease class selectivity in the first three network layers.** (a) Test accuracy (y-axis) as a function of $\alpha$ (x-axis) when class selectivity regularization is restricted to the first three network layers in ResNet18 trained on Tiny ImageNet. (b) Test accuracy (y-axis) as a function of mean class selectivity in the first three layers (x-axis) in ResNet18 trained on Tiny ImageNet. (c) and (d) are identical to (a) and (b), respectively, but for ResNet20 trained on CIFAR10. Error bars = 95% confidence intervals.

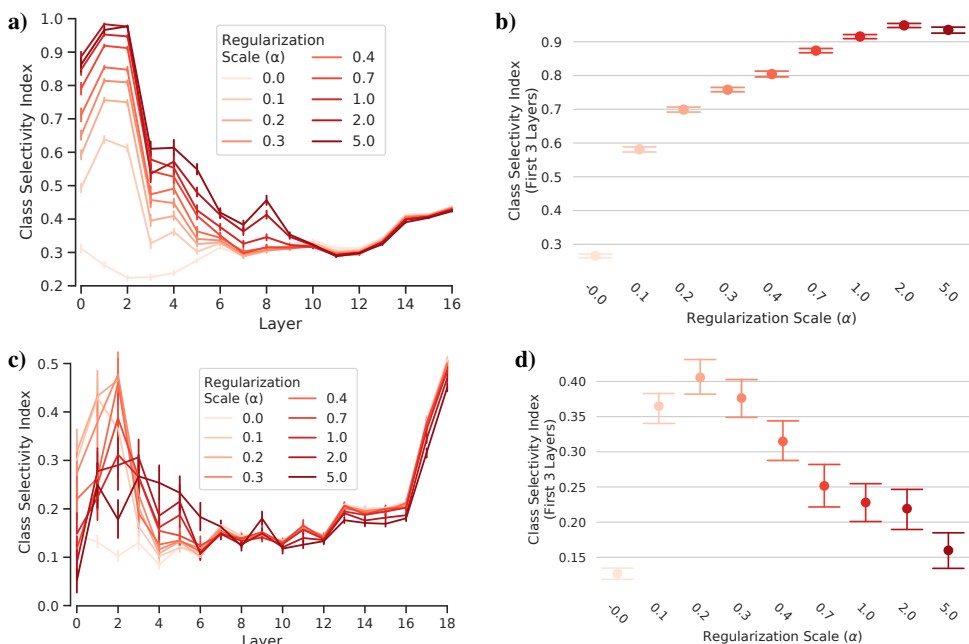

**Figure A22: Regularizing to increase class selectivity in the first three network layers.** (a) Mean class selectivity (y-axis) as a function of layer (x-axis) for different values of $\alpha$ (intensity of red) when class selectivity regularization is restricted to the first three network layers in ResNet18 trained on Tiny ImageNet. (b) Mean class selectivity in the first three layers (y-axis) as a function of $\alpha$ (x-axis) in ResNet18 trained on Tiny ImageNet. (c) and (d) are identical to (a) and (b), respectively, but for ResNet20 trained on CIFAR10. Error bars = 95% confidence intervals.

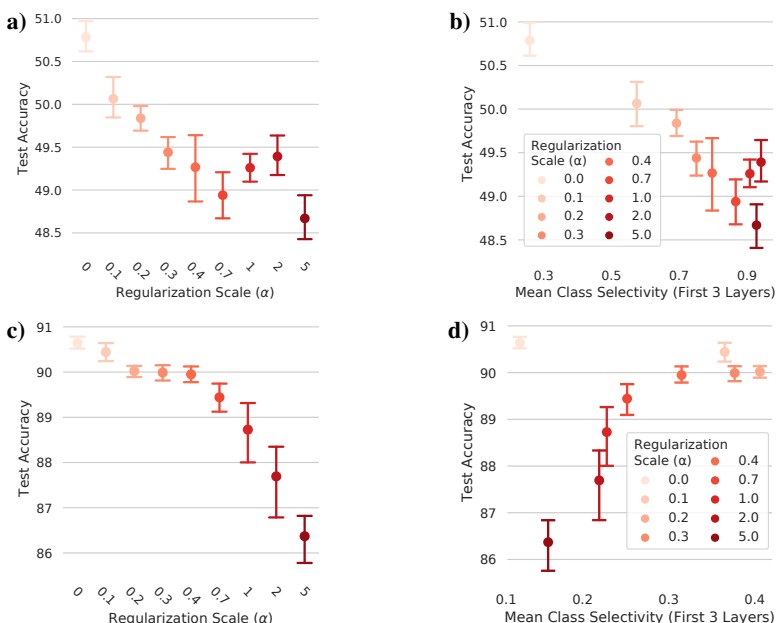

**Figure A23: Effects on test accuracy when regularizing to increase class selectivity in the first three network layers.** (a) Test accuracy (y-axis) as a function of $\alpha$ (x-axis) when class selectivity regularization is restricted to the first three network layers in ResNet18 trained on Tiny ImageNet. (b) Test accuracy (y-axis) as a function of mean class selectivity in the first three layers (x-axis) in ResNet18 trained on Tiny ImageNet. (c) and (d) are identical to (a) and (b), respectively, but for ResNet20 trained on CIFAR10. Error bars = 95% confidence intervals.

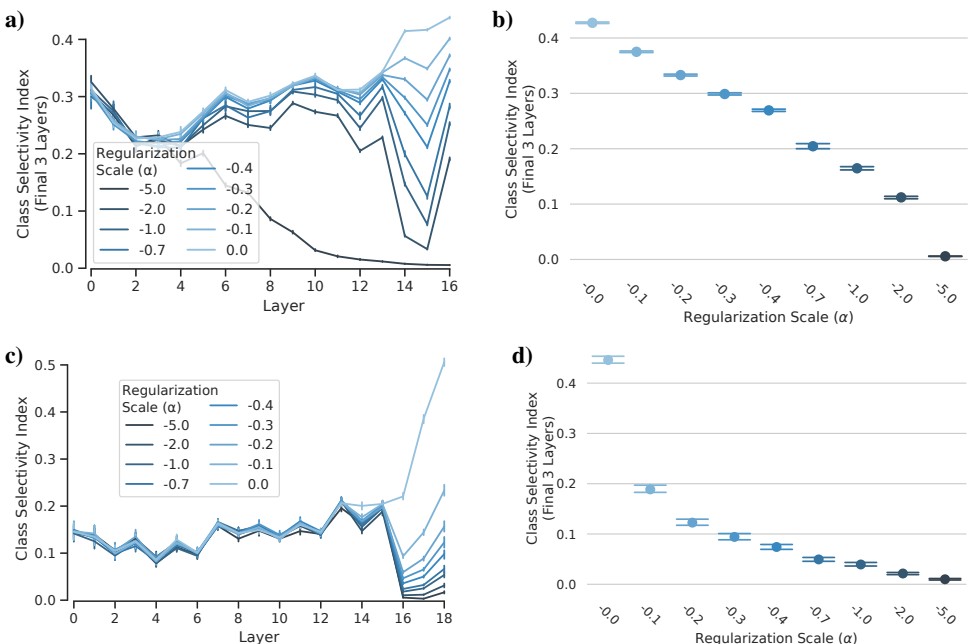

**Figure A24: Regularizing to decrease class selectivity in the last three network layers.** (**a**) Mean class selectivity (y-axis) as a function of layer (x-axis) for different values of $\alpha$ (intensity of blue) when class selectivity regularization is restricted to the last three network layers in ResNet18 trained on Tiny ImageNet. (**b**) Mean class selectivity in the last three layers (y-axis) as a function of $\alpha$ (x-axis) in ResNet18 trained on Tiny ImageNet. (**c**) and (**d**) are identical to (**a**) and (**b**), respectively, but for ResNet20 trained on CIFAR10. Error bars = 95% confidence intervals.

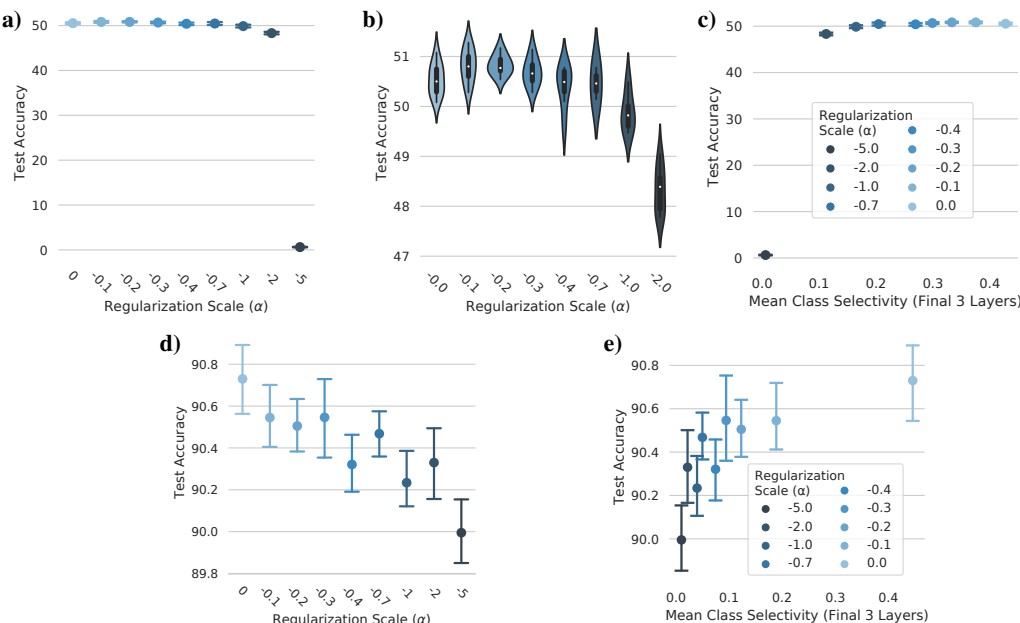

**Figure A25: Effects on test accuracy when regularizing to decrease class selectivity in the last three network layers.** (**a**) Test accuracy (y-axis) as a function of $\alpha$ (x-axis) when class selectivity regularization is restricted to the last three network layers in ResNet18 trained on Tiny ImageNet. (**b**) Similar to (**a**), but for a subset of $\alpha$ values. (**c**) Test accuracy (y-axis) as a function of mean class selectivity in the last three layers (x-axis) in ResNet18 trained on Tiny ImageNet. (**d**) and (**e**) are identical to (**a**) and (**c**), respectively, but for ResNet20 trained on CIFAR10. Error bars = 95% confidence intervals.

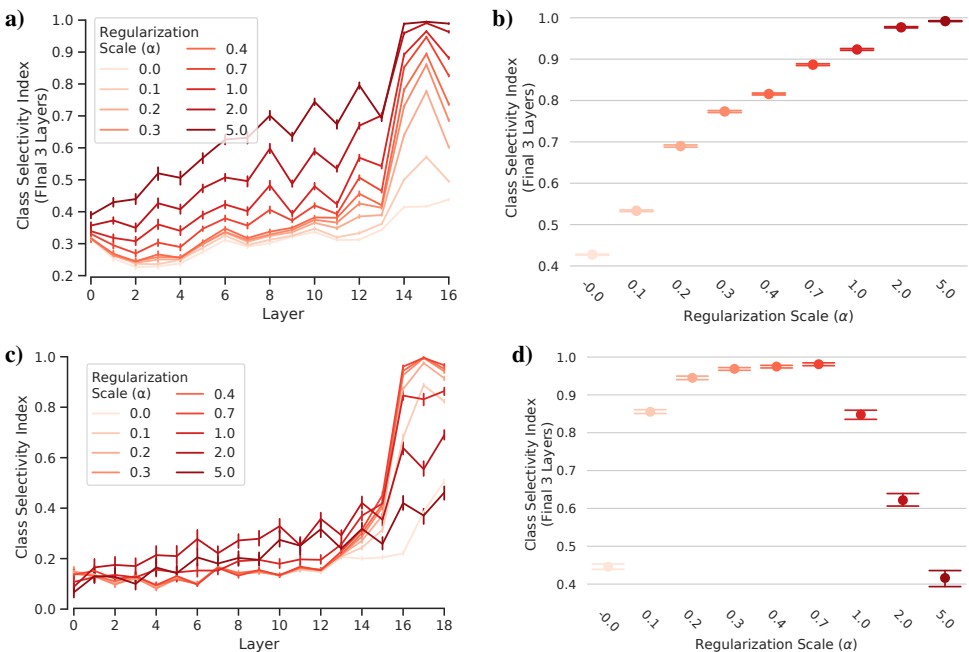

**Figure A26: Regularizing to increase class selectivity in the last three network layers.** (**a**) Mean class selectivity (y-axis) as a function of layer (x-axis) for different values of $\alpha$ (intensity of red) when class selectivity regularization is restricted to the last three network layers in ResNet18 trained on Tiny ImageNet. (**b**) Mean class selectivity in the last three layers (y-axis) as a function of $\alpha$ (x-axis) in ResNet18 trained on Tiny ImageNet. (**c**) and (**d**) are identical to (**a**) and (**b**), respectively, but for ResNet20 trained on CIFAR10. Error bars = 95% confidence intervals.

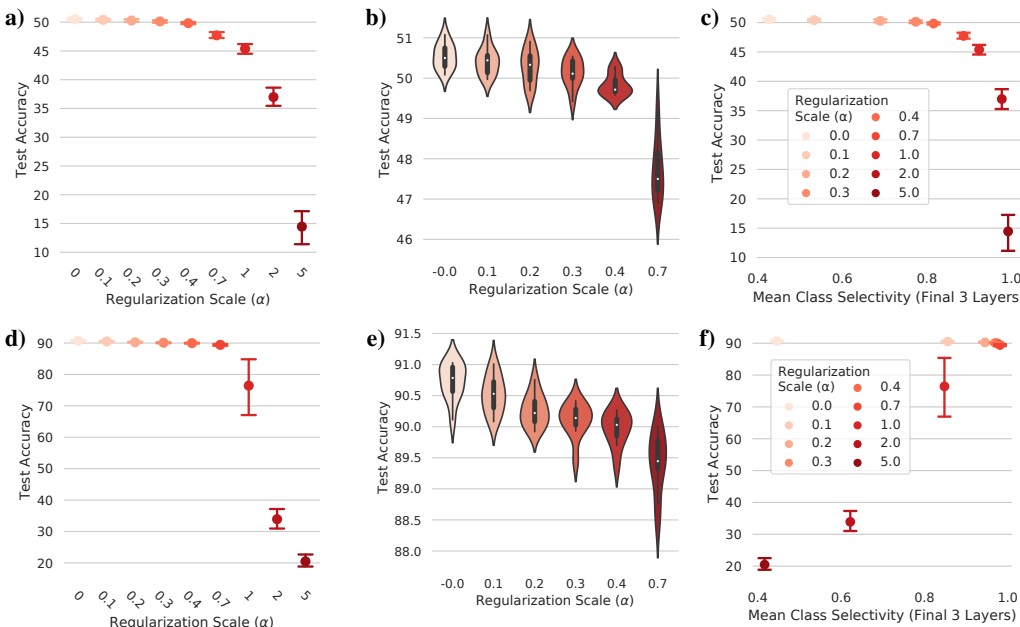

**Figure A27: Effects on test accuracy when regularizing to increase class selectivity in the last three network layers.** (**a**) Test accuracy (y-axis) as a function of $\alpha$ (x-axis) when class selectivity regularization is restricted to the last three network layers in ResNet18 trained on Tiny ImageNet. (**b**) Similar to (**a**), but for a subset of $\alpha$ values. (**c**) Test accuracy (y-axis) as a function of mean class selectivity in the last three layers (x-axis) in ResNet18 trained on Tiny ImageNet. (**d-f**) are identical to (**a-c**), respectively, but for ResNet20 trained on CIFAR10. Error bars = 95% confidence intervals.

### A.16 AN INABILITY TO CHANGE PREFERRED CLASSES DURING TRAINING DOES NOT EXPLAIN SELECTIVITY-INDUCED PERFORMANCE DEFICITS

One potential limitation of our selectivity regularizer is that regularizing to increase class selectivity could discourage individual neurons from changing their preferred class during training; units would be locked in to the class they preferred at initialization. If changing preferred classes is necessary to improve test accuracy, then regularizing to increase class selectivity could impose a constraint on performance. We controlled for this possibility in two ways. First, we analyzed the statistics of preferred class changes that occur as a function of the class selectivity regularization scale. We examined the proportion of units that change classes at least once during training when $\alpha \geq 0$. The mean proportion is one, for all examined regularization scales and models (Figure A28), indicating that regularizing to increase selectivity does not lock units into their initial preferred class. We also examined whether regularizing to increase selectivity affects the number of times a unit changes its preferred class during training. In ResNet18 trained on Tiny ImageNet, we found that the mean number of preferred class changes decreases as a function of $\alpha$ for $\alpha \geq 0.3$ (Figure A29a). However, we note that the number of preferred class changes is roughly equal for $\alpha = \{0, 0.1, 0.2\}$, even though test accuracy for $\alpha = \{0.1, 0.2\}$ is significantly lower than for $\alpha = 0$, indicating that preferred class changes cannot fully account for decreased test accuracy. Furthermore, in ResNet20 trained on CIFAR10 there is no clear relationship between $\alpha$ and the number of preferred class changes (Figure A29b). These results indicate that the test accuracy impairment caused by regularizing to increase class selectivity cannot be fully explained by the regularizer preventing units from changing their preferred class during training.

We also controlled for the possibility of preferred class "lock-in" by warming up selectivity regularization during training, which would allow units' preferred classes to change during the early period of training, when learning-induced changes are most significant (Achille et al., 2018; Gur-Ari et al., 2018; Frankle et al., 2020). We implemented selectivity regularization warmup in an analogous manner to learning rate warmup—by scaling $\alpha$ (see Equation 2) linearly over the interval [0,1] across the epochs first five epochs of training. We observed qualitatively similar results as when we did not warm up selectivity regularization (Figure A30, Figure A31). Regularizing to reduce class selectivity either has minimal negative effects (ResNet20 trained on CIFAR10) or improves test accuracy (ResNet18 trained on Tiny ImageNet), while regularizing to increase class selectivity impairs test accuracy in all examined models (Figure A31). Interestingly, the severity of the test accuracy impairment is slightly reduced when using regularization warmup. This indicates that some of the test accuracy deficit from regularizing to increase class selectivity is attributable to the regularizer forcing the network into suboptimal solutions early in training. Nevertheless, the test accuracy deficit imparted by increased class selectivity remains even when the class selectivity regularizer is warmed up, further supporting the claim that that class selectivity is not sufficient for network performance.

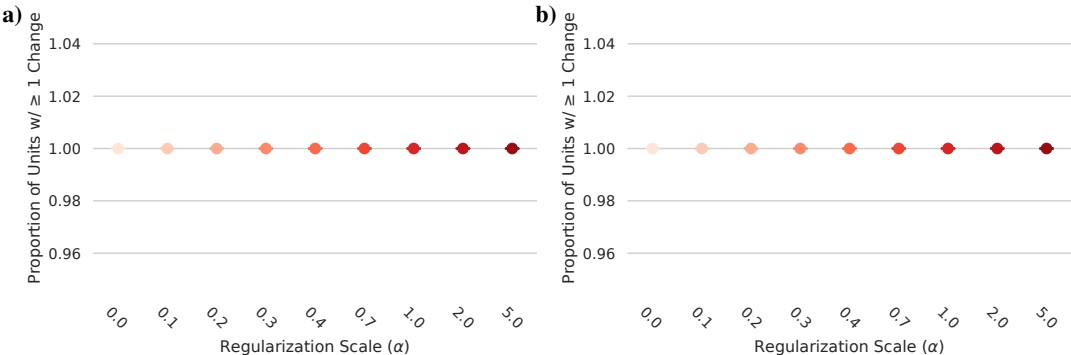

**Figure A28: Units change preferred classes during training even when regularizing to increase class selectivity.** (**a**) Mean proportion of units that change preferred classes at least once during training (y-axis) as a function of regularization scale (x-axis; hue) for ResNet18 trained on Tiny ImageNet. (**b**) Identical to (**a**), but for ResNet20 trained on CIFAR10. Error bars = 95% confidence intervals.

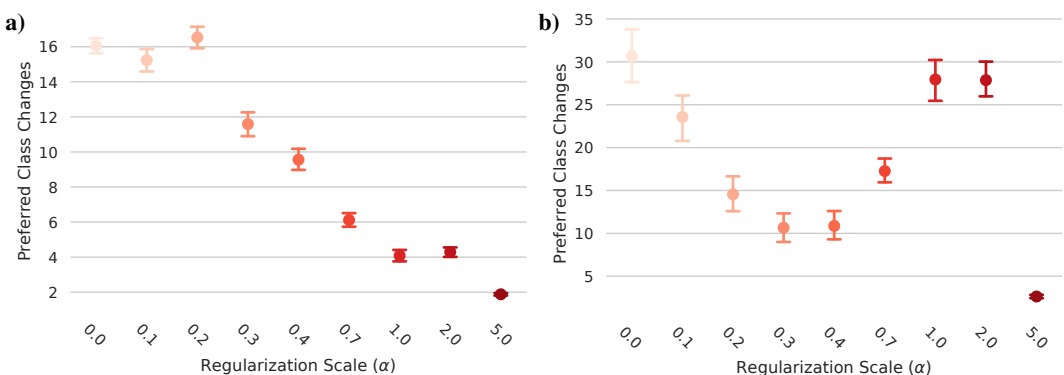

**Figure A29: Regularizing to increase class selectivity has inconsistent effects on the number of times units change their preferred class during training.** (a) Mean number of preferred class changes across units (y-axis) as a function of regularization scale (x-axis; hue) for ResNet18 trained on Tiny ImageNet. (b) Identical to (a), but for ResNet20 trained on CIFAR10. Error bars = 95% confidence intervals.

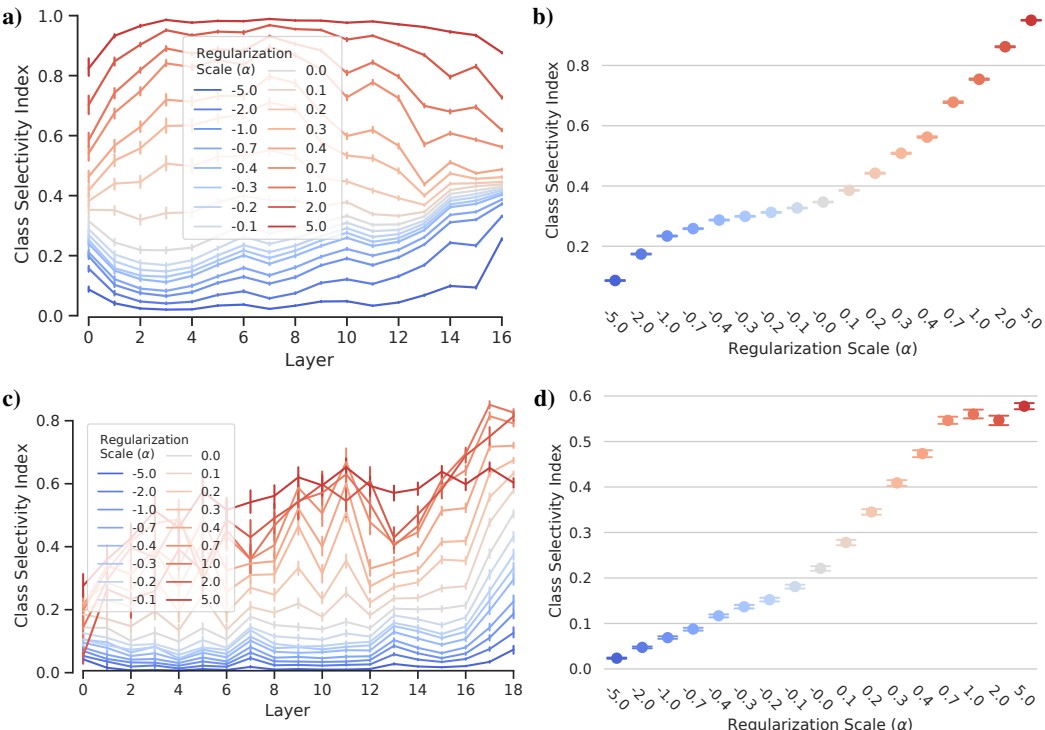

**Figure A30: Effects of warming up class selectivity regularization.** (a) Mean class selectivity (y-axis) as a function of layer (x-axis) for different values of $\alpha$ (hue) when class selectivity regularization is warmed up over the first 5 epochs in ResNet18 trained on Tiny ImageNet. (b) Mean class selectivity across all layers (y-axis) as a function of $\alpha$ (x-axis) when warming up selectivity regularization in ResNet18 trained on Tiny ImageNet. (c) and (d) are identical to (a) and (b), respectively, but for ResNet20 trained on CIFAR10. Error bars = 95% confidence intervals.

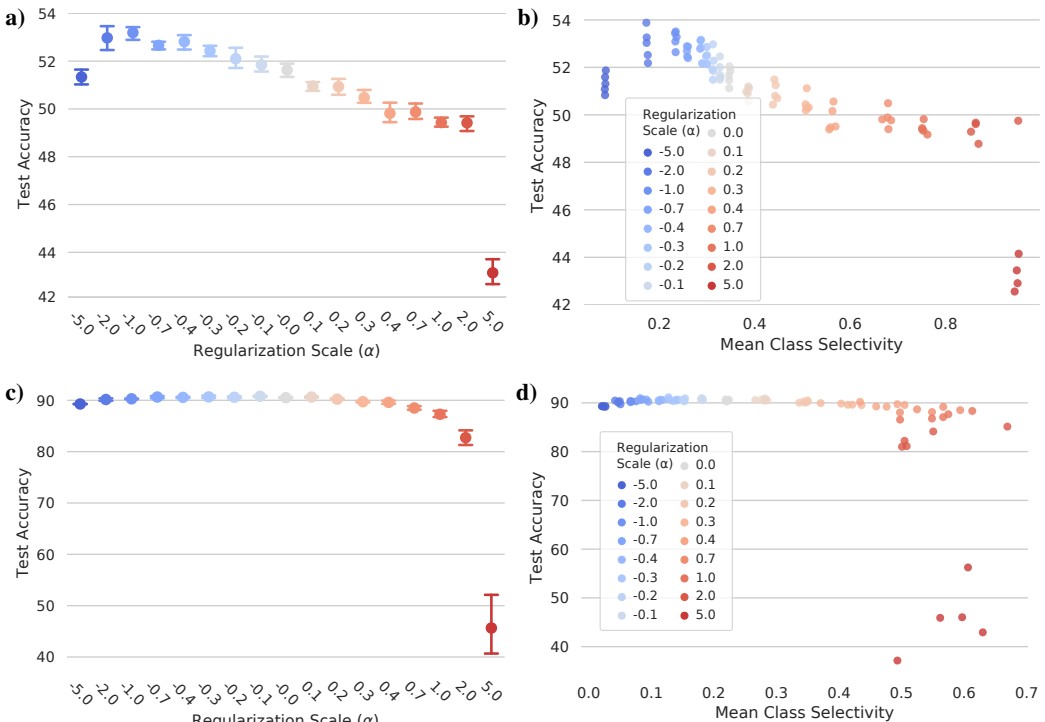

**Figure A31: Effects of class selectivity regularization warmup on test accuracy.** (a) Test accuracy (y-axis) as a function of class selectivity regularization scale ($\alpha$; x-axis and hue) when class selectivity regularization is warmed up over the first 5 epochs in ResNet18 trained on Tiny ImageNet. (b) Test accuracy (y-axis) as a function of mean class selectivity (x-axis) when warming up selectivity regularization in ResNet18 trained on Tiny ImageNet. Each data point denotes a single network. (c) and (d) are identical to (a) and (b), respectively, but for ResNet20 trained on CIFAR10. Error bars = 95% confidence intervals.

## A.17 DIRECTLY COMPARING THE EFFECTS OF REGULARIZING TO INCREASE VS. DECREASE CLASS SELECTIVITY

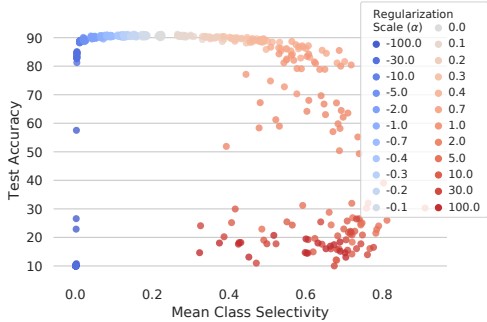

**Figure A32: Increasing class selectivity has rapid and deleterious effects on test accuracy compared to reducing class selectivity in ResNet20 trained on CIFAR10.** Test accuracy (y-axis) as a function of mean class selectivity (x-axis). Each data point denotes a network.

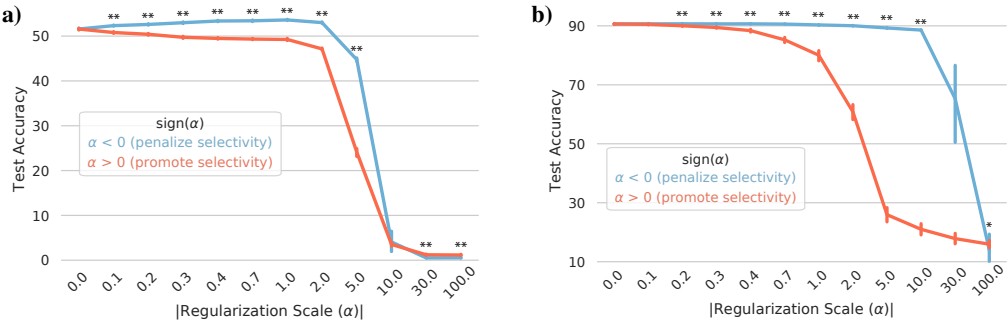

**Figure A33: Directly comparing promoting vs penalizing class selectivity.** (**a**) Test accuracy (y-axis) as a function of regularization scale ($\alpha$; x-axis) when promoting ($\alpha > 0$, red) or penalizing ($\alpha < 0$, blue) class selectivity in ResNet18 trained on Tiny Imagenet. **$p < 2 \times 10^{-5}$ difference between penalizing vs. promoting selectivity, Wilcoxon rank-sum test, Bonferroni-corrected. (**b**) same as (**a**) but for ResNet20 trained on CIFAR10. *$p < 0.05$, **$p < 6 \times 10^{-6}$ difference, Wilcoxon rank-sum test, Bonferroni-corrected. Error bars denote bootstrapped 95% confidence intervals.

### A.18 EFFECTS OF CLASS SELECTIVITY REGULARIZATION IN RESNET50 TRAINED ON TINY IMAGENET

We also examined the effect regularizing to control class selectivity in ResNet50 trained on Tiny ImageNet. Our results were qualitatively similar to those observed in ResNet18: class selectivity in trained networks correlates with $\alpha$ (Figure A34a), and regularizing to decrease class selectivity can improve test accuracy, while regularizing to increase class selectivity impairs test accuracy (Figure A34b).

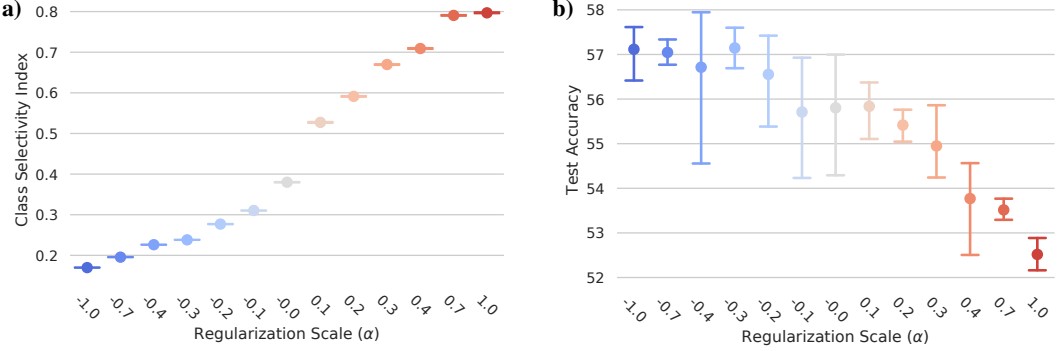

**Figure A34: Effects of class selectivity regularization in ResNet50 trained on Tiny ImageNet** (**a**) Mean class selectivity (y-axis) as a function of regularization scale ($\alpha$; x-axis) when promoting ($\alpha > 0$, red) or penalizing ($\alpha < 0$, blue) class selectivity. (**b**) Test accuracy (y-axis) as a function of $\alpha$ (x-axis). Note that the increased size of the 95% confidence intervals is at least partially-attributable to training only 5 replicates per $\alpha$ compared to the 20 replicates per $\alpha$ used for other models.

