# OpenReview forum: "Selectivity considered harmful: evaluating the causal impact of class selectivity in DNNs"
_ICLR.cc/2021/Conference — ICLR 2021 Poster_

### Official Review · AnonReviewer3 · 2020-10-26
**Interesting paper with a problematic methodology**

**Rating:** 6
**Confidence:** 3

**Review:**

[Update after authors' reply]

In light of the authors' reply, I have updated my review to favor acceptance. I appreciate the additional experiments. It will be up to the readers to determine how to interpret those additional results.

---
The authors attempt to ascertain whether single-neuron class selectivity is beneficial or harmful to overall network performance. They do this by adding a regularization term to the training loss, which measures how selective network units are. They observe that discouraging selectivity can have a small benefit, and generally doesn't harm performance even at very high values. Conversely, encouraging selectivity dramatically decreases performance. They perform additional analyses, e.g. checking that selectivity is not simply masked by linearly dividing it across units.

The paper is interesting, well-written, and the experiments are thorough  (minor quibble: I would have liked a bit more emphasis on layer-by-layer analyses, which only seem to occur in the last few figures of the appendix?).

My main problem is with the regularizer (Eq. 1). in my (possibly flawed) understanding, it seems to have additional effects, that are not discussed in the paper (unless I have missed it) and complicate the interpretation of results.

IIUC, ascending the regularizer value does not simply increase "selectivity" in general, but *current* selectivity. That is, it encourages the neuron to increase its preference for whatever it's preferring *right now*, and discourages / prevents it from ever switching preferences, even if that might be beneficial to overall network training.

Thus, it is possible that the dramatic drop in performance is not due simply to increased selectivity in general, but to locking the preferences of individual units early on, when they are essentially random and likely sub-optimal.

It is not obvious how to control for such a possibility. At the very least, one might check how often units switch preference over training, with and without the regularizer. Perhaps only turning on the regularizer when switches become rare (if they ever do!) might help?

It seems that this problem does not occur in the reverse direction: decreasing selectivity does not seem to lock up a specific preference, but rather to reduce preference in general, which is the expected interpretation (IIUC). As such, the results for negative alpha seem unaffected.

Again, I may have missed something in the paper. Otherwise, the problem needs to be at the very least discussed, and the authors should find some way to estimate it and possibly control for it, for the paper to be accepted.

---

> ### Author Response · Authors · 2020-11-19
> **Response to Reviewer 3**
>
> We’re glad you found the paper interesting, well-written, and thorough.
>
> >I would have liked a bit more emphasis on layer-by-layer analyses, which only seem to occur in the last few figures of the appendix
>
> We also find the layer-by-layer analyses interesting, but unfortunately had to move them to the appendix given the space limitations in the main text.
>
> >Ascending the regularizer value does not simply increase "selectivity" in general, but current selectivity. That is, it encourages the neuron to increase its preference for whatever it's preferring right now, and discourages / prevents it from ever switching preferences, even if that might be beneficial to overall network training. Thus, it is possible that the dramatic drop in performance is not due simply to increased selectivity in general, but to locking the preferences of individual units early on, when they are essentially random and likely sub-optimal.
>
> Your understanding of the regularizer is indeed accurate, and this is indeed a valid concern and an insightful observation. We addressed it in multiple ways, all of which confirm that this concern cannot account for deleterious impact of regularizing to increase selectivity. Please note that the information here is also contained in the newly-added Appendix A16, and summarized in the main text (Section 4.3, paragraph 4).
>
> We first examined the proportion of units that change classes at least once during training when alpha ≥ 0 (Appendix A16, paragraph 1). The mean proportion is one, for all examined regularization scales and models (Figure A28), indicating that regularizing to increase selectivity does not lock units into their initial preferred class.
>
> We also examined whether regularizing to increase selectivity affects the number of times a unit changes its preferred class during training (Appendix A16, paragraph 1). In ResNet18 trained on Tiny ImageNet, we found that the mean number of preferred class changes decreases as a function of alpha for alpha ≥ 0.3 (Figure A29a). However, the number of preferred class changes is roughly equal for alpha = {0, 0.1, 0.2}, even though test accuracy for alpha = {0.1, 0.2} is significantly lower than for alpha = 0, indicating that preferred class changes cannot fully account for decreased test accuracy. Most strikingly, even for alpha = 5.0, units still change their preferred class ~2 times over training, with extremely low variance across units. Furthermore, in ResNet20 trained on CIFAR10 there is no straightforward relationship between alpha and the number of preferred class changes (Figure A29b). These results indicate that the test accuracy impairment caused by regularizing to increase class selectivity cannot be fully explained by the regularizer preventing units from changing their preferred class during training.
>
> We further controlled for the possibility of preferred class "lock-in" by warming up selectivity regularization during training, which would allow units' preferred classes to change freely during the early period of training, when learning-induced changes are most significant (Achille et al., 2018; Gur-Ari et al., 2018, Frankle et al., 2020). We implemented selectivity regularization warmup in an analogous manner to learning rate warmup—by scaling alpha linearly over the interval [0,1] across the epochs first five epochs of training. We observed qualitatively similar results as when we did not warm up selectivity regularization (Figure A30, Figure A31). Regularizing to reduce class selectivity either has minimal negative effects (ResNet20 trained on CIFAR10) or improves test accuracy (ResNet18 trained on Tiny ImageNet), while regularizing to increase class selectivity impairs test accuracy in all examined models (Figure A31). Interestingly, the severity of the test accuracy impairment is slightly reduced when using regularization warmup. This indicates that some of the test accuracy deficit from regularizing to increase class selectivity is attributable to the regularizer forcing the network into suboptimal solutions early in training. Nevertheless, the test accuracy deficit imparted by increased class selectivity remains substantial even when the class selectivity regularizer is warmed up, indicating that test accuracy deficit cannot be fully explained by preferred class lock-in.
>
> We hope these three analyses mitigate your concerns about regularizing to increase class selectivity.

---

### Official Review · AnonReviewer4 · 2020-10-28
**Class selectivity in individual units is not critical for CNN classification performance**

**Rating:** 6
**Confidence:** 4

**Review:**

This paper examines the impact of forcing units in a CNN to be more or less “class-selective” – i.e. respond preferentially to one image class compared to another.  The approach taken is to include a regularizer in the loss that directly penalizes or encourages class selectivity in individual units. They report that penalizing class selectivity at intermediate layers has little-to-no effect on classification performance, and in some cases mildly improves performance. They authors conclude that class selectivity is not an essential component of successful performance in CNNs, and that methods which use class selectivity to interpret CNNs should be approached with caution.

--

Pros:

The authors address a basic/important question – whether class selectivity in individual units of a deep network is related to task performance on a classification task.

The writing is overall clear. I found the motivation, experiments, and results easy to understand.

The analyses are fairly extensive.

Overall, I found the results compelling.

--

Limitations and questions:

The authors results show that class selectivity, as they define it, is not critical to good performance, in the sense that networks can perform well without showing class selectivity. However, this fact does not demonstrate that class selectivity is unimportant to a network trained in the normal way without regularization. It is of course possible that networks trained with different losses could learn different strategies for solving the same task. -- Update: In my opinion, this fact is not highlighted prominently enough, including in the revised version. --

I didn’t find the off-axis selectivity analyses very convincing. Simply using CCA to show that the regularization has an effect on the network subspace doesn’t strike me as very convincing, since it’s unclear how the subspace differs. It seems like the key question is whether there are linear projections that exhibit class selectivity, which could be addressed in a more straightforward fashion using linear classifiers, trained and tested in the usual way.

Related to the above point, I would have liked to see a measure of class selectivity that is more directly related to discrimination – i.e. the ability to discriminate exemplars of one class from another. A natural choice in my opinion would be a measure like d-prime that measures the within vs. between class distance. How would the results look if they used linear discriminant analysis to project the activations of each layer onto a low-dimensional subspace that maximized class discriminability and then measured the overall within vs. between class separation in this space? -- Update: The authors have repeated their analysis with d-prime. They do not observe an improvement when regularizing against d-prime, but report an asymmetry, where promoting class selectivity leads to a greater decrement than down-weighting class selectivity --

One of the ways that class selectivity has been used is to examine selectivity for objects in a network trained to recognize scenes. This paper does not examine this type of selectivity for sub-classes. Does scene recognition require first classifying objects?

---

My rating remains unchanged. The authors have addressed some but not all of the issues raised, as noted above.

---

> ### Author Response · Authors · 2020-11-19
> **Response to Reviewer 4**
>
> We appreciate that you consider the questions we are pursuing to be fundamental and important, and that you found our writing clear and our analyses extensive. You raised a number of interesting concerns that we attempted to comprehensively address through a number of new experiments and analyses. Also please note that we have broken our response up over two comments due to length limitations.
>
> >The authors results show that class selectivity, as they define it, is not critical to good performance, in the sense that networks can perform well without showing class selectivity. However, this fact does not demonstrate that class selectivity is unimportant to a network trained in the normal way without regularization. It is of course possible that networks trained with different losses could learn different strategies for solving the same task. This fact should be highlighted more prominently in my opinion.
>
> We completely agree with your conclusions. Indeed, this is one of the main motivations for our experiments: prior studies examining the importance of class selectivity in networks trained the “normal way” (i.e. in the absence of selectivity regularization) have yielded conflicting results (Morcos et al., 2018b; Kanda et al., 2020; Amjad et al., 2018; Zhou et al., 2018; Dalvi et al., 2019; Donnelly and Roegiest, 2019). We tried to emphasize the difference between our approach and “standard” methods in the introduction, but as per your suggestion we have added a section to the conclusion that reiterates this idea.
>
> We also believe our results speak directly to your second point (different losses causing networks to learn different solutions): Our CCA analyses found that increasing the strength of regularization against class selectivity caused hidden-layer representations to become more dissimilar from those of networks trained without selectivity regularization (Figure 2, Section 4.2; Figure A3). This indicates that selectivity regularization does indeed cause networks to learn different solutions compared to unregularized networks, and that the degree of difference is proportional to the degree of regularization.
>
> >I didn’t find the off-axis selectivity analyses very convincing. Simply using CCA to show that the regularization has an effect on the network subspace doesn’t strike me as very convincing, since it’s unclear how the subspace differs. The upper-bound analysis is more to-the-point, but I didn’t understand why the projection was constrained to be a rotation (orthonormal). It seems like the key question is whether there are linear projections that exhibit class selectivity, which could be addressed in a more straightforward fashion using linear classifiers, trained and tested in the usual way.
>
> We agree that there is no silver bullet analysis solution for confirming that selectivity is not simply shifted off of unit-aligned axes, which is why we performed two complementary analyses. As you point out, the CCA measure does not provide information about _how_ representations differ. But the intent of the analyses were to verify that the regularizer does not just shift selectivity off-axis via rotation or some other affine transformation, a conclusion that requires only a relative comparison: whether the values of the metrics change as a function of class selectivity regularization intensity. And we do indeed observe this effect (Figure 2, Section 4.2; Figures A3-A4, Sections A6-A7).
>
> The reason we constrained the upper-bound projection to be orthonormal was because the non-orthonormal solution to maximizing selectivity is degenerate: project all axes onto the single direction in activation space with the highest class selectivity. We regret not making this explicit, and have added this explanation to the description of the upper bound estimation method in the appendix.

---

> > ### Author Response · Authors · 2020-11-19
> > **Response to Reviewer 4, cont'd**
> >
> > >Related to the above point, I would have liked to see a measure of class selectivity that is more directly related to discrimination – i.e. the ability to discriminate exemplars of one class from another. A natural choice in my opinion would be a measure like d-prime that measures the within vs. between class distance. How would the results look if they used linear discriminant analysis to project the activations of each layer onto a low-dimensional subspace that maximized class discriminability and then measured the overall within vs. between class separation in this space?
> >
> > This is an interesting idea! Because d’ is differentiable with respect to the model parameters, we addressed this using the same regularization approach to control the d’ learned by individual units (Appendix A14). Our procedure was identical to class selectivity regularization, except instead of calculating SI for each unit, we calculated d’. When regularizing d’, we found that d’ correlates strongly with SI across all values of alpha (Spearman's rho = 0.83, Resnet18 trained on Tiny ImageNet; rho = 0.90 ResNet20 trained on CIFAR10; p < 10e-5 for both). Although regularizing against d’ did not improve test accuracy, we found similar overall results when regularizing to control d’ as we did when regularizing to control class selectivity: the negative impact of regularizing to decrease d’ is far less dramatic than the negative impact of regularizing to increase d’ (Figure A19). The asymmetry in effect between increasing vs. decreasing d' is therefore consistent with the effect of increasing vs. decreasing class selectivity, indicating that neither class selectivity nor discriminability are sufficient nor strictly necessary for CNN performance. We added a section (Appendix A14) that details these findings.
> >
> > >One of the ways that class selectivity has been used is to examine selectivity for objects in a network trained to recognize scenes. This paper does not examine this type of selectivity for sub-classes. Does scene recognition require first classifying objects?
> >
> > This is an interesting question, although it’s challenging to answer using our paradigm, as it requires manipulating the object selectivity of units in a network trained to perform scene recognition. We could, however, more easily answer a different, though related question: how does class selectivity regularization affect selectivity for features at different levels of abstraction? Specifically, we could use the Broden dataset (Bau et al., 2017), which combines several densely-labelled datasets in one that includes a broad range of objects, scenes, object parts, textures, materials, and colors in a range of contexts. Bau et al. assembled this dataset to investigate selectivity for “concepts” at different levels of abstraction (“color” is an example of a low-level concept, while “object” is an example of a high-level concept). We aimed to include analyses using this dataset in our revised manuscript, but unfortunately adapting this dataset to our paradigm is non-trivial, and we were unable to finish the analyses in time for our response. We would, however, include them in the updated version of the paper.

---

### Official Review · AnonReviewer2 · 2020-10-29
**an interesting story about whether selectivity is necessary at intermediate representations**

**Rating:** 7
**Confidence:** 4

**Review:**

This paper asks the interesting question of whether you need individual neuron (or even population level) class selectivity at intermediate stages in order to have good classification performance. The authors introduce a regularization term to the loss that controls the amount of selectivity in the units of the network. They find that the selectivity of the units in standard networks can be reduced while maintaining classification performance.

Overall, this paper presents a compelling story. Although it could be strengthened by increasing the clarity in some of the presented results, the experimental results seem detailed and rigorous leading to a recommended accept.

Positives:
* The question of whether individual units or populations of units are the right domain to study is currently topical for both the machine learning community and the neuroscience community.
* The question of necessity of selectivity is similarly topical for both ML and neuroscience.
* The paper presents a nice step back from much of the single unit interpretability literature, highlighting that it is unclear what some of these results mean if the selective units are not actually necessary for the task performance.
* The experimental results (other than the optimization concerns below) seem sound, particularly including the multiple replicates + error bars to verify the experimental findings.

Concerns and Suggestions:
* The stability of optimization when including the regularization scale is somewhat suspicious, specifically at high values of $|\alpha|$ where there is not a direct relationship between the test accuracy and the observed class selectivity (ie figure 4b). If optimization is working correctly, wouldn’t one suspect that at high $\alpha$ the mean class selectivity should be close to 1, while at low $\alpha$ the mean class selectivity should be close to 0? Instead the mean accuracy seems to behave erratically, suggesting a potential instability in the optimization. Although these values are perhaps uninteresting, it raises the concern that there could be an interaction happening at intermediate values.
* Related to the above, the paper may be strengthened if the main text results were from the leaky ReLU models that are presented in the appendix, as it seems like the story is the same but the confound of individual unit selectivity is removed.
* The authors note that, by definition, the final layer must be class selective in a classification task (which means that all networks studied have class selective units). It thus seems like the question that the authors are addressing is whether selectivity in *intermediate* units is beneficial for selectivity in downstream units (rather than investigating whether selectivity is generally helpful). This should be clarified in the introduction.
* The investigation using CCA to test whether the selectivity is simply rotated off axis is quite compelling, however providing some intuition as to how a network could use non-selective features in order to perform a task necessitating selectivity downstream would strengthen the story.

---

> ### Author Response · Authors · 2020-11-19
> **Response to Reviewer 2**
>
> Thank you for describing our work as topical, necessary, and experimentally sound. Your comments seem largely addressable through clarifications which we implemented in the manuscript and articulate here, and we performed additional experiments and analyses in order to resolve your remaining concerns. Also please note that we have broken our response up over two comments due to length limitations.
>
> >The stability of optimization when including the regularization scale is somewhat suspicious, specifically at high values of |α| where there is not a direct relationship between the test accuracy and the observed class selectivity (ie figure 4b). If optimization is working correctly, wouldn’t one suspect that at high α the mean class selectivity should be close to 1, while at low α the accuracy should be close to 0? Instead the mean accuracy seems to behave erratically, suggesting a potential instability in the optimization. Although these values are perhaps uninteresting, it raises the concern that there could be an interaction happening at intermediate values.
>
> Please note that our reply assumes that you intended to say “class selectivity” in place of “accuracy” in this comment. At extreme values of α (e.g. |α| >= 10), the selectivity regularizer likely dominates the loss and scales it into a regime that precludes good solutions, while at intermediate values the regularizer appears to be well-behaved. A number of factors support this interpretation. First, the behavior of the class selectivity regularizer is consistent across different models—ResNet18, ResNet20, VGG16—and selectivity metrics—class selectivity index, precision (Appendix A.13), mutual information (Appendix A.13), d’ (Appendix A.14). We introduced the mutual information and d’ analyses in order to help assuage your concerns, as well as those of Reviewer 4.
>
> Across all examined models and metrics, selectivity changes consistently with the sign and magnitude of intermediate values of α, and collapses at extreme values of α. The results for d’ are particularly noteworthy, as they were obtained by regularizing to control d’ directly, demonstrating that our findings hold even when regularizing different metrics of class information. And across all examined models and metrics, extreme α values seem to yield degenerate solutions. One particularly striking example is the finding that increasing α causes an increase in the proportion of dead units in ResNet20 (as noted in your review), and that controlling for this via leaky-ReLUs does not rescue network performance (Appendix A.10).
>
> In an attempt to further verify that the regularizer does not push the network into degenerate solutions at intermediate values of α (and to address reviewer 3’s concerns regarding class selectivity being “locked in” at initialization), we used an approach analogous to learning rate warmup, but for the selectivity regularizer: we slowly scaled alpha from zero to a target α over the first five epochs of training (Appendix A.16). This minimizes the effects of the selectivity regularizer during the early period of training, when learning-induced changes are most significant (Achille et al., 2018; Gur-Ari et al., 2018, Frankle et al., 2020). The results when using regularizer warmup are extremely qualitatively similar to when using a constant α, indicating that the selectivity regularizer does not push the network into degenerate or edge-case solutions at intermediate values of α.
>
> >Related to the above, the paper may be strengthened if the main text results were from the leaky ReLU models that are presented in the appendix, as it seems like the story is the same but the confound of individual unit selectivity is removed.
>
> ResNet18 trained on Tiny ImageNet, which the main text focuses on, did not have an issue with dead units. We have made this finding more explicit by adding Figure A7, which shows that the proportion of dead units in ResNet18 stays at or near zero, even for large values of alpha, and mentioning it in the text (Section 4.3, last sentence of paragraph 3). Accordingly, we only performed leaky-ReLU controls for ResNet20 trained on CIFAR10. Given that the main text focuses on the results for ResNet18 trained on Tiny ImageNet and that space is limited, we thought it made sense to keep the ResNet20 leaky-ReLU results in the appendix. We apologize for the lack of clarity.

---

> > ### Author Response · Authors · 2020-11-19
> > **Response to Reviewer 2, cont'd**
> >
> > >The authors note that, by definition, the final layer must be class selective in a classification task (which means that all networks studied have class selective units). It thus seems like the question that the authors are addressing is whether selectivity in intermediate units is beneficial for selectivity in downstream units (rather than investigating whether selectivity is generally helpful). This should be clarified in the introduction.
> >
> > We regret using ambiguous terminology: by “output layer” and “final layer” we meant the logits, not the final hidden layer. Every mention of the “output” or “final” layer is now clarified with the term “logit” (Section 3, paragraph 2; Section 4.3, paragraph 4; Appendix A.1, paragraph 4). To your point, it could be possible to perform classification with very weakly-selective units in the final hidden layer (see our response to your next comment), though our experiments indicate that some selectivity in the final/later hidden layers appears beneficial (see the Appendix A.4 and Appendix A.15).
> >
> > >The investigation using CCA to test whether the selectivity is simply rotated off axis is quite compelling, however providing some intuition as to how a network could use non-selective features in order to perform a task necessitating selectivity downstream would strengthen the story.
> >
> > Even when regularizing very strongly to decrease selectivity, there are units (predominantly in deeper layers) that have non-zero selectivity. Even so, the presence of non-selective units in a network trained to perform image classification is indeed counterintuitive. It may be difficult to envision how non-selective units could shape representations in a manner that is useful for a classification task. One possibility is that the class-conditional joint distribution of activations across units facilitates readout. Put another way, the correlations between units' activations can help separate the distributions of activations for different classes. There is evidence that correlated variability between neurons can facilitate information readout in the brain (Zylberberg et al., 2016; Leavitt et al., 2017; Zylberberg, 2018; Nogueira, 2020). The toy diagram in Figure 6a of Leavitt et al. (2017) provides a visual intuition for this scenario, and Zylberberg (2018) develops a theoretical grounding for it (though both use the vernacular of neuroscience). This explanation and these citations have been added to the manuscript. Given the tight timeline, we focused on experiments and analyses in this response, but we could add a diagram similar to Leavitt et al. (2017) if you think it would improve clarity and help build intuition. We have also added a paragraph to the discussion section that conveys the same intuition as our response here.

---

> > > ### Comment · AnonReviewer2 · 2020-11-24
> > > **response to authors**
> > >
> > > Thank you for all of the clarifications, the updated experiments, and interesting discussion with how to interpret non-selective units.
> > >
> > > (and yes -- I did mean to say "class selectivity", it is now updated in my comment).

---

### Author Response · Authors · 2020-11-19
**General response to reviewers**

We appreciate that all three reviewers praised the strengths of our work. It is gratifying to see the motivation for our work explicitly affirmed: “The authors address a basic/important question” (R4); “The question of necessity of selectivity is similarly topical for both ML and neuroscience”. (R2). We were also happy to hear that “The experimental results...seem sound” (R2), “The analyses are fairly extensive” (R4), and “the experiments are thorough” (R3). R4 “found the results compelling”, and the context and implications of our findings are highlighted by R2, who stated: “The paper presents a nice step back from much of the single unit interpretability literature, highlighting that it is unclear what some of these results mean if the selective units are not actually necessary for the task performance.”

We also appreciate the thoroughness and thoughtfulness of the reviewers’ concerns and suggestions. We have carried out extensive additional experiments and analyses in an attempt to thoroughly resolve all reviewer concerns that could not be addressed through clarification alone, and updated the manuscript accordingly. We summarize the new experiments and analyses here, and describe them in detail in our responses to individual reviewers.

First, we replicated our experiments with warm-up applied to the selectivity regularizer (i.e. linearly scaling from zero to $\alpha$) over the first five training epochs in order to mitigate the effects of the regularizer during the critical early period of training (Appendix A16). Regularizer warmup did not alter the results or conclusions of our work, which addresses concerns raised by two reviewers: R2, who expressed concern about the regularizer destabilizing the optimization process, and R3, who expressed concern that regularizing to increase class selectivity could undermine network performance by preventing units from changing their preferred class during training (so-called “class lock-in”). We further addressed R3’s concerns about class lock-in by analyzing the effect of selectivity regularization on units’ preferred class changes over the course of training, and did not find conclusive evidence that lock-in can fully account for decreased test accuracy.

R4 inquired about the use of a measure of class selectivity that is more directly related to discriminability (e.g. $d’$). Because $d’$ is differentiable with respect to the model parameters, we applied our regularization technique using $d’$ in place of class selectivity and found qualitatively similar results when regularizing to control d’ as when regularizing to control class selectivity (Appendix A14). Finally, we replicated our results in ResNet50 trained on Tiny ImageNet (Appendix A15); we note that the range of tested $\alpha$ are smaller, and confidence intervals are larger because we only used 5 replicates per alpha instead of 20, both due to time constraints). While none of the reviewers explicitly requested this, we feel it bolsters confidence in the consistency of our findings and the relevance and appeal of our work.

Thank you,

Authors

---

### Comment · ~Naomi_Saphra1 · 2021-05-25
**Releasing code?**

Will you be releasing a repository for this paper?

---

### Decision · Program_Chairs · 2021-01-07
**Final Decision**

**Decision:**

Accept (Poster)

**Comment:**

This paper has received three positive reviews. In general, the reviewers have commented on the importance of the question related to how much selectivity is needed from units of a neural network for good classification -- from both the neuroscience and ML perspectives. The reviewers also commented on the thoroughness of the experiments and the general readability of the paper. This paper should be accepted if possible.